# Impairments of spatial memory in an Alzheimer's disease model via degeneration of hippocampal cholinergic synapses

Houze Zhu[1,2], Huanhuan Yan[1,2], Na Tang[1,2], Xinyan Li[1,2], Pei Pang[1,2], Hao Li[1,2], Wenting Chen[1,2], Yu Guo[1,2], Shu Shu[1,2], You Cai[1,2], Lei Pei[2,3], Dan Liu[2,4], Min-Hua Luo [5], Hengye Man [2,6], Qing Tian[2,7], Yangling Mu[1,2], Ling-Qiang Zhu[1,2] & Youming Lu[1,2]

Choline acetyltransferase neurons in the vertical diagonal band of Broca (vChATs) degenerate in the early stage of Alzheimer's disease (AD). Here, we report that vChATs directly innervate newly generated immature neurons (NGIs) in the dorsal hippocampus (dNGIs) of adult mice and regulate both the dNGIs survival and spatial pattern separation. In a mouse model that exhibits amyloid-β plaques similar to AD patients, cholinergic synaptic transmission, dNGI survival and spatial pattern separation are impaired. Activation of vChATs with theta burst stimulation (TBS) that alleviates the decay in cholinergic synaptic transmission effectively protects against spatial pattern separation impairments in the AD mice and this protection was completely abolished by inhibiting the dNGIs survival. Thus, the impairments of pattern separation-associated spatial memory in AD mice are in part caused by degeneration of cholinergic synaptic transmission that modulates the dNGIs survival.

[1] Department of Physiology, School of Basic Medicine and Tongji Medical College, Huazhong University of Science and Technology, Wuhan 4030030, China. [2] The Institute for Brain Research, Collaborative Innovation Center for Brain Science, Huazhong University of Science and Technology, Wuhan 430030, China. [3] Department of Neurobiology, School of Basic Medicine and Tongji Medical College, Huazhong University of Science and Technology, Wuhan 4030030, China. [4] Department of Genetics, School of Basic Medicine and Tongji Medical College, Huazhong University of Science and Technology, Wuhan 4030030, China. [5] State Key Laboratory of Virology, CAS Center for Excellence in Brain Science and Intelligence Technology (CEBSIT), Wuhan Institute of Virology, Chinese Academy of Sciences, Wuhan 430071, China. [6] Department of Biology, Boston University, 5 Cummington St, Boston, MA 02215, USA. [7] Department of Pathophysiology, School of Basic Medicine and Tongji Medical College, Huazhong University of Science and Technology, Wuhan 430030, China. Houze Zhu, Huanhuan Yan, Na Tang and Xinyan Li contributed equally to this work. Correspondence and requests for materials should be addressed to Y.M. (email: ymu@hust.edu.cn) or to L.-Q.Z. (email: zhulq@hust.edu.cn) or to Y.L. (email: lym@hust.edu.cn)

Deposition of senile plaques that primarily consist of amyloid-β (Aβ) is a major pathological hallmark in the brains of Alzheimer's disease (AD) and has long been considered to be associated with a progressive loss of central neurons[1–5]. However, recent studies indicate that spatial memory loss that is known as an early clinical sign of AD is due to synaptic dysfunction rather than neuronal death. In AD patients, the impairments of spatial memory correlate with a reduction of excitatory glutamatergic terminals[6, 7]. In Tg2576-APPswe mice (AD mice) that carry a transgene encoding the 695-amino-acid isoform of the human Aβ precursor protein with the Swedish mutation and exhibit plaque pathologies similar to those in AD patients[8], synaptic loss in the CA1 hippocampus reduces the capability of spatial information acquisition[9, 10]. However it is still unknown which of many thousands of synapses in the brain undergo degeneration in the early stage of AD and whether this selective degeneration contributes directly to spatial memory loss.

Acetylcholine (ACh) modifies neuronal excitability, alters pre-synaptic neurotransmitter release and coordinates the firing of groups of neurons[11–13]. In the hippocampus, ACh is released from axon terminals of choline acetyltransferase neurons (ChATs) in the vertical diagonal band of Broca (vDB) (vChATs) and plays a role in a range of cognitive activities, such as attention, learning and memory and consciousness[14–17]. However the hippocampus consists of diverse types of neuronal cells, including excitatory neurons and GABAergic inhibitory neurons, which of these cell types establish a direct synaptic connection with vChATs remain unknown and a role of this direct cholinergic synaptic connection in spatial learning and memory has not been previously investigated.

To map the specific neuronal cells that develop synaptic connections with vChATs in adult mice, we used a genetically modified Cre-dependent anterograde monosynaptic tracing system. We demonstrated that vChATs directly innervate newly generated immature neurons (NGIs) in the dorsal zone of the hippocampus (dNGIs) of adult mice. In AD mice, cholinergic synaptic transmission is impaired and this impairment contributes to the loss of pattern separation-dependent spatial memory.

## Results

**vChATs directly innervate dNGIs**. We used ChATs-Cre[GFP+/+] mice, in which Cre-enhanced green fluorescence protein (GFP) is expressed under the control of the ChAT promoter (Fig. 1a, b). Staining the sections with an antibody against ChAT confirmed that Cre-GFP was expressed in ChAT neurons (Fig. 1b). A high titer (0.2 μl, $6 \times 10^{10}$ genomic particles/ml) of monosynaptic anterograde herpes simplex virus (HSV) vector that encoded a double-floxed inverted open reading frame mCherry (HSV-DIO-mCherry virus) was subsequently injected into the vDB region of the ChATs-Cre[GFP+/+] mice. At 3 days after the injection, the brain sections were processed. A bright red fluorescent signal (mCherry) was detected in the GFP-positive vChATs (GFP[+]mCherry[+]) and their direct targeting (postsynaptic) neurons in the dorsal dentate gyrus (dDG) of the adult mice (Fig. 1c). In the dDG, mCherry was exclusively expressed in a group of granule cells that were predominately located in the inner one-third of the granule cell layer (Fig. 1c). Most of these mCherry[+] cells expressed doublecortin (DCX, mCherry[+]DCX[+], Fig. 1d; Supplementary Fig. 1a). DCX is widely established as a marker of immature neurons[18], and mCherry[+]DCX[+] cells were therefore classified as newly generated immature neurons (NGIs) in the dDG region (dNGIs).

To further determine a direct synaptic connection between vChATs and dNGIs, we used a genetically modified retrograde monosynaptic tracing system (Fig. 1e). We created mutant mouse lines with an inducible expression of the avian viral receptor TVA and rabies G-GFP in dNGIs of adult mice (dNGIs[TVA/G+/+] mice) by crossing TVA/G[loxP/loxP] mice with the Nestin-Cre[ER] mice. Following oral administration of tamoxifen (TAM, 100 mg in corn oil/kg body weight), TVA/G-GFP were expressed in dNGIs (Fig. 1f). We subsequently injected a high titer (1.5 μl of $3 \times 10^9$ genomic particles/ml) of the synaptic retrograde ΔG-rabies viruses that encoded mCherry into the dorsal hippocampus of the dNGIs[TVA/G+/+] mice. At 3 days after the injection, a mCherry signal (mCherry[+]) was detected in GFP-positive dNGIs (GFP[+]/mCherry[+], Fig. 1g) and their direct presynaptic vChATs (mCherry[+]ChATs[+], Fig. 1h; Supplementary Fig. 1b) and a few of neurons in the entorhinal cortex (Supplementary Fig. 1c). Altogether, these findings demonstrate that vChATs directly innervate dNGIs in adult mice.

To determine whether a direct synaptic connection between vChATs and dNGIs is functional, we engineered vChATs with the expression of channelrhodopsin-2-E123A (ChR2, Fig. 2a). We created ΔG-rabies viruses that encoded the double-floxed inverted open reading frame of ChR2-GFP (ΔG-rabies-DIO-ChR2). A high titer (1 μl, $2 \times 10^{11}$ genomic particles/ml) of virus particles was injected into the dDG of dNGIs[TVA/G+/+] mice, resulting in the expression of ChR2 in dNGIs and their direct presynaptic vChATs (ChATs[ChR2+] mice, Fig. 2a; Supplementary Fig. 2). The successful expression of functional ChR2 channels in vChATs in vivo was evidenced by whole-cell patch clamp recordings showing reliable neuronal activation with consistent firing patterns in vChAT[ChR2+] neurons from the vChAT[ChR2+] mice in response to illumination with blue laser lights (Fig. 2a).

To record cholinergic synaptic transmission between vChATs and dNGIs, brain slices were prepared from the vChAT[ChR2+] mice at 7, 10, 16, 22 and 28 days after the injection (DAI) of ΔG-rabies-DIO-ChR2 virus particles (Fig. 2b). Whole-cell patch clamp recordings were performed from GFP[+] dNGIs (Fig. 2c). The illumination of vChATs[ChR2+] neurons evoked cholinergic synaptic currents in dNGIs at 10 DAI. The currents peaked in the dNGIs at 16 DAI (mean amplitude, 103.2 ± 9.1 pA; mean ± SEM; $n = 28$ cells/6 mice) and decreased 6 days later (Fig. 2d), suggesting that vChATs establish the functional synaptic connections with dNGIs only, but not with the matured granule cells in the dorsal hippocampus. The currents that were evoked by illumination of vChATs were sensitive to the general muscarinic acetylcholine receptor antagonist atropine (5 μM) and a selective type-1 muscarinic (M1) acetylcholine receptor antagonist VU0255035 (5 μM, Fig. 2e), but not to the nicotinic acetylcholine receptor inhibitor (Ni) and the type-2 (M2), type-3 (M3) or type-4 (M4) of the muscarinic acetylcholine receptor inhibitors, showing that M1 receptor mediates cholinergic synaptic transmission in dNGIs. The currents were fitted with single exponentials (Fig. 2f) with the averaged latency of 3.27 ± 0.28 ms (Fig. 2g), which is in line with monosynaptic transmission.

**Cholinergic transmission regulates the dNGIs survival**. NGIs in adult mice are derived from endogenous neural progenitor cells (NPCs) in the sub-granular zone[19, 20]; and these cells may proliferate and differentiate into new excitatory granule cells (NGCs) that extend axonal projections and functionally integrate into the existing circuits via the formation of excitatory synapses with their targets[21, 22]. The generation of NGCs or adult neurogenesis is a unique mode of structural plasticity that plays a role in spatial learning and memory[23–25]. Our finding of a direct synaptic connection between vChATs and dNGIs indicated that vChATs may control the growth of dNGIs in adult mice.

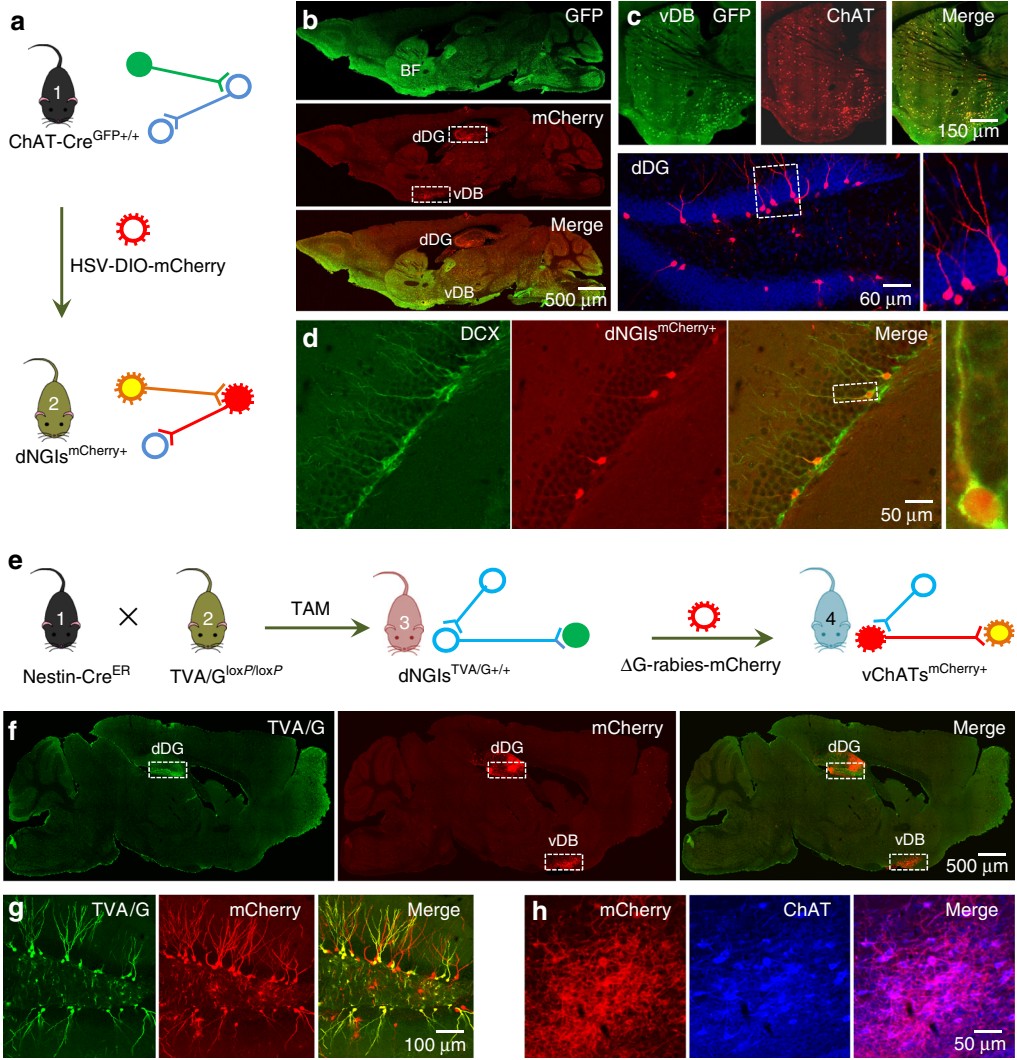

**Fig. 1** vChATs directly innervate dNGIs. **a** Monosynaptic anterograde tracing strategy shows the application of HSV-DIO-mCherry virus in ChAT-Cre^GFP+/+ mice for labeling postsynaptic cells (red) of vChATs (yellow). **b** A brain section (top) from a ChATs-Cre^GFP+/+ mouse shows GFP expression (green, top). 3 days after the injection of HSV-DIO-mCherry virus (0.2 μl), mCherry-expressing cells were detected in both the vDB (top) and dDG (red, middle and bottom) regions of ChAT-Cre^GFP+/+ mice. **c** GFP-expressing ChATs (green) in the section stained with anti-ChAT (red) in the vDB region (top). The mCherry-expressing cells (red) in the sub-granular zone of the dDG region stained with DAPI (blue, bottom). **d** The mCherry-expressing cells in the dDG region are co-labeled with anti-DCX (blue). **e** Monosynaptic retrograde tracing strategy shows that NGI^TVA/G+/+ mice were generated by crossing Nestin-Cre^ER mice with TVA/G^loxP/loxP mice. Following the administration of tamoxifen (TAM), ΔG-rabies-mCherry virus particles (red) were injected into the dDG region of the NGI^TVA/G+/+ mice for labeling presynaptic cells (red) of dNGIs (yellow). **f** A brain section reveals the TVA/G-GFP cells (green) in the dDG region and mCherry in both the dDG and vDB of a dNGI^TVA/G+/+ mouse 3 days after injection of the ΔG-rabies-mCherry virus particles (1.5 μl). **g** TVA/G^+/+ cells (green) in the dDG region were co-labeled with mCherry from the dNGI^TVA/G+/+ mice 3 days after injection of the ΔG-rabies-mCherry virus. **h** The mCherry^+ cells (red) in the vDB region were co-labeled with anti-ChAT (blue) from the dNGI^TVA/G+/+ mice 3 days after injection of the ΔG-rabies-mCherry virus. **a**–**h**, similar results were seen in each of five experiments

To investigate this possibility, we selectively activated vChATs by delivering a blue laser light with a theta burst (TBS) paradigm through an optical fiber (0.2 mm diameter) located in the vDB region of the vChAT^ChR2+ mice once per day for 16 consecutive days (16-day TBS). This protocol was implemented because it is beneficial for synaptic transmission and plasticity in the hippocampus[26]. We subsequently dated the adult-born neurons in the vChATs^ChR2+ mice by injecting 5-bromo-2-deoxyuridine (BrdU) and following their fate (Fig. 3a). The vChAT^ChR2+ mice were administered a single dose (230 mg/kg bodyweight) of BrdU and euthanized at 1, 3, 7, 11, 21, and 28 days after the BrdU injection (DAI) to enable quantification before and after the critical period of the dNGIs survival (Fig. 3b). The number of BrdU-positive cells (BrdU^+)

increased from 1 to 7 DAI and subsequently decreased between 11 and 28 DAI; the rise curves that represent endogenous progenitor cell (EPC) proliferation were identical between the groups (Fig. 3c; Supplementary Fig. 3). The magnitude of the decrease, which reflects the dNGI survival[27], was significantly decreased in the vChATs^ChR2+ mice compared with controls (Fig. 3c). These data indicate that TBS treatment increases the survival of dNGIs in adult mice. The number of NGCs in the vChATs^ChR2+ mice, as identified by the co-expression of GFP and NeuN (GFP^+ NeuN^+ cells; Fig. 3d) at 28 DAI, was 2-fold higher than the control mice (mean ± SEM, $n = 11$ mice/group; Fig. 3e). These gain-of-function studies demonstrate that activation of vChATs with TBS enhances dNGIs survival in adult mice.

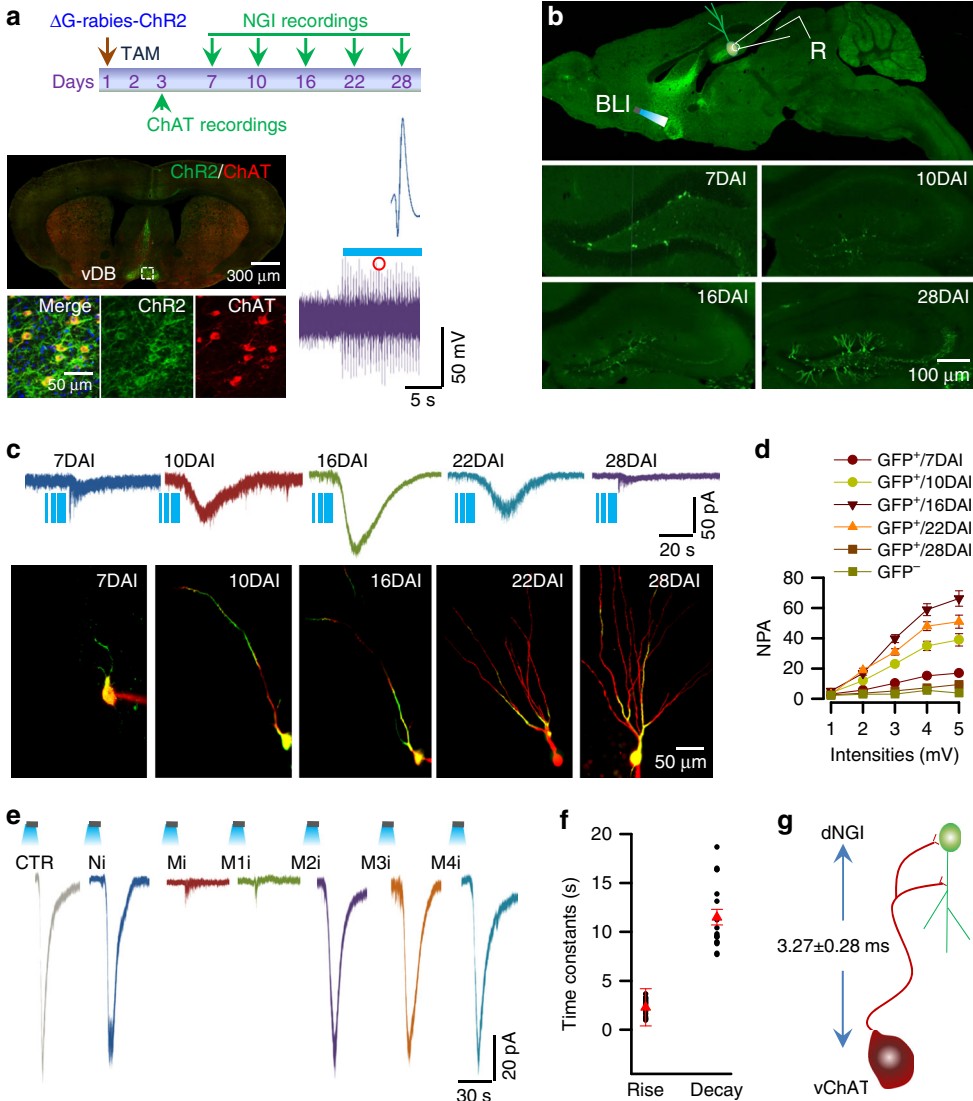

**Fig. 2** vChATs functionally innervate dNGIs. **a** Experimental schedule (top) shows the generation of vChATs$^{ChR2+}$ mice by injecting the ΔG-rabies-ChR2-GFP virus into the dDG region of NGIs$^{TVA/G+/+}$ mice. Whole-cell recordings were performed from vChATs$^{ChR2+}$ (day 3) and dNGIs$^{GFP+}$ cells (from days 7 to 28) in slices from the vChATs$^{ChR2+}$ mice. A slice (left bottom) from a vChATs$^{ChR2+}$ mouse reveals ChR2-expressing cells (green) stained with anti-ChAT (red). Representative traces show the in vivo recording of a single (blue trace) and a cluster (purple) of action potentials in a vChATs$^{ChR2+}$ mouse in response to blue laser light at 10 Hz. **b** Experimental arrangements (top) show a recording electrode (R) and an optical fiber for light illumination (BLI) and representative images (bottom) show dNGIs (green) in the dDG region of vChATs$^{ChR2+}$ mice at 7, 10, 16, or 28 days after the ΔG-rabies-ChR2-GFP virus injection (DAI). **c** Whole-cell recordings in the individual dNGIs (green, bottom) that were filled with Lucifer yellow (red) through the recording electrodes in the vChATs$^{ChR2+}$ mice at 7, 10, 16, 22 or 28 days after the ΔG-rabies-ChR2-GFP virus injection (DAI). Representative recordings (top) from the individual dNGIs evoked by the illumination of vChATs. **d** Averaged normalized peak amplitudes (NPA) of the currents vs. light intensities from 1 to 5 mV/mm$^2$ are plotted (mean ± SEM, n = 28 cells/6 mice/group). **e** Representative currents in a dNGI at 16 DAI were sensitive to a muscarinic receptor inhibitor (Mi, atropine 5 μM), or a selective type-1 muscarinic receptor inhibitor VU0255035 (M1i), but not to either the nicotinic receptor inhibitor (Ni, 10 μM tubocurarine) or to the other types of muscarinic receptor inhibitors (5 μM AF-DX116 for M2 receptor; DAU5884 for M3, and PD102807 for M4 receptor). **f** The actual (black circles) and the averaged values (red triangles) of rise and decay time constants are plotted (mean ± SEM, n = 18 cells/9 mice). **g** Latencies of synaptic responses evoked by illuminating vChATs were 3.27 ± 0.28 ms (mean ± SEM, n = 18 cells/9 mice)

To determine the specificity of cholinergic activation in the dNGIs survival, we stimulated excitatory synapses of pyramidal neurons in the entorhinal cortex (EC$_{PN}$). We used the EC$_{PN}$$^{ChR2+}$ mice, in which ChR2 was expressed in excitatory pyramidal neurons. TBS was delivered through an optical fiber located in the entorhinal cortical layer II region of the EC$_{PN}$$^{ChR2+}$ mice once per day for 16 consecutive days. BrdU$^+$ cells in the dorsal hippocampus were then examined. Our data showed that TBS activation of EC$_{PN}$ did not alter the survival of dNGIs in adult mice (Supplementary Fig. 4a–d).

We subsequently demonstrated that cholinergic synaptic transmission between vChATs and dNGIs is essential for the survival of dNGIs. We generated ΔG-rabies-DIO-NpHR virus particles (2.8 × 10$^{11}$ genomic particles/ml), which were subsequently injected into the dorsal hippocampus of the dNGIs$^{TVA/G}$$^{+/+}$ mice, resulting in the expression of halorhodopsin (NpHR) in vChATs (vChATs$^{NpHR+}$ mice). NpHR is a light-sensitive chloride channel that hyperpolarizes neuronal cells. Three days after the NpHR viral injection, amber laser light (ALL) was delivered directly onto the vChATs of the vChATs$^{NpHR+}$ mice through an

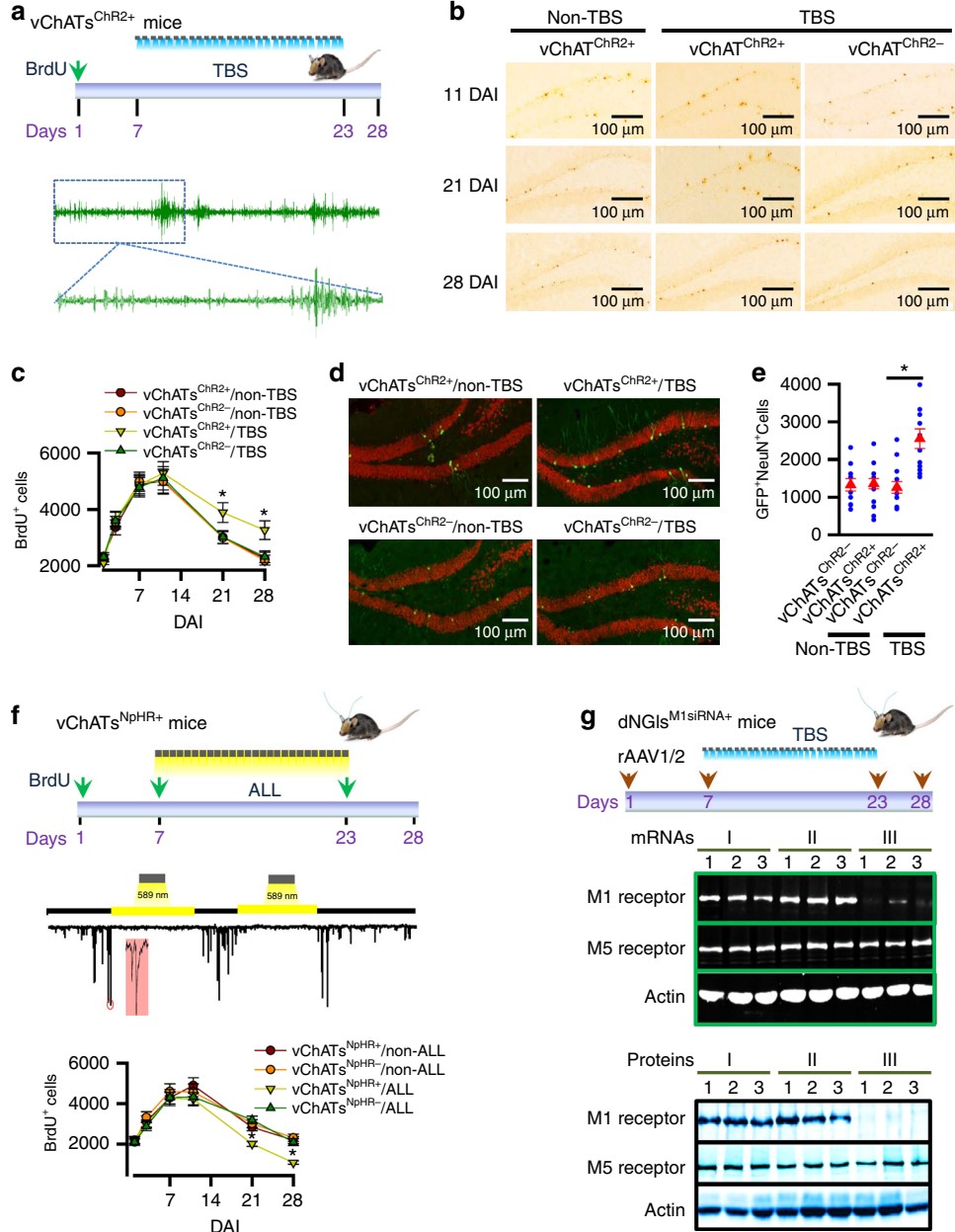

**Fig. 3** Cholinergic transmission regulates the dNGIs survival. **a** Experimental schedule shows that mice at $120 \pm 2$ days old of age were administered with a single dose of BrdU. 7 days after the BrdU administration (DAI), TBS was applied for 16 consecutive days. BrdU staining was performed at 1, 3, 7, 11, 21, or 28DAI. Representative theta waves recorded from dDG region of vChATs$^{ChR2+}$ mice in response to TBS. **b**, **c** Representative images **b**, and summarized numbers **c**, of BrdU$^+$ cells in the vChATs$^{ChR2+}$ and vChATs$^{ChR2-}$ mice treated without (non-TBS) or with TBS (mean $\pm$ SEM, $n = 9$ mice/group, $*p < 0.01$, two-way ANOVA with Bonferroni post-hoc test). **d**, **e** Representative images **d** and a plot **e** shows the actual (blue circles) and the averaged numbers (red triangles) of GFP$^+$BrdU$^+$ cells in the vChATs$^{ChR2+}$ and vChATs$^{ChR2-}$ mice treated without (non-TBS) or with TBS (mean $\pm$SEM, $n = 11$ mice/group, $*p < 0.01$, two-way ANOVA with Bonferroni post-hoc test). **f** Experimental schedule (top) for the BrdU administration and ALL application in both vChATs$^{NpHR+}$ and vChATs$^{NpHR-}$ mice in vivo. BrdU staining was performed at 1, 3, 7, 11, 21, or 28DAI. Representative recordings (middle) indicate that the application of ALL (yellow light) blocked spontaneous excitatory currents in vChATs$^{NpHR+}$ cells at a holding potential of $-60$ mV. Bar graph (bottom) indicates the numbers of BrdU$^+$ cells in the vChATs$^{NpHR+}$ and vChATs$^{NpHR-}$ mice treated without (non-ALL) or with ALL (ALL). BrdU labeling was performed at 1, 3, 7, 11, 21, or 28 DAI (mean $\pm$ SEM, $n = 9$ mice/group, $*p < 0.01$, two-way ANOVA with Bonferroni post-hoc test). **g** Experimental arrangements show the schedules of rAAV1/2 virus injection and BrdU administration. 7 days after the injection, TBS was applied for 16 consecutive days. Western blots and RT-PCR analysis were performed 5 days after the end of TBS. Representative images of mRNAs and proteins show 3 replicates (lanes 1, 2 and 3) in each condition; no virus (I), sM1siRNA (II) and M1siRNA virus injections (III). Similar results were seen in each of the four experiments

optic fiber (0.2 mm diameter) once per day (10 Hz for 15 min/day) for 16 consecutive days (Fig. 3f). To determine the effects of vChATs inactivation on dNGIs survival, the vChATs$^{NpHR+}$ mice were injected with a single dose (230 mg/kg body weight) of BrdU. 7 days after the injection (DAI), mice were treated with 16-

days ALL. Mice were euthanized at 1, 3, 7, 11, 21 or 28 DAI. The numbers of BrdU$^+$ cells significantly decreased at both 21 and 28 DAI (Fig. 3f), whereas the numbers of proliferating glial cells, as identified by the expression of GFAP (GFAP$^+$), were unchanged in the vChATs$^{NpHR+}$ mice (Supplementary Fig. 5) compared with

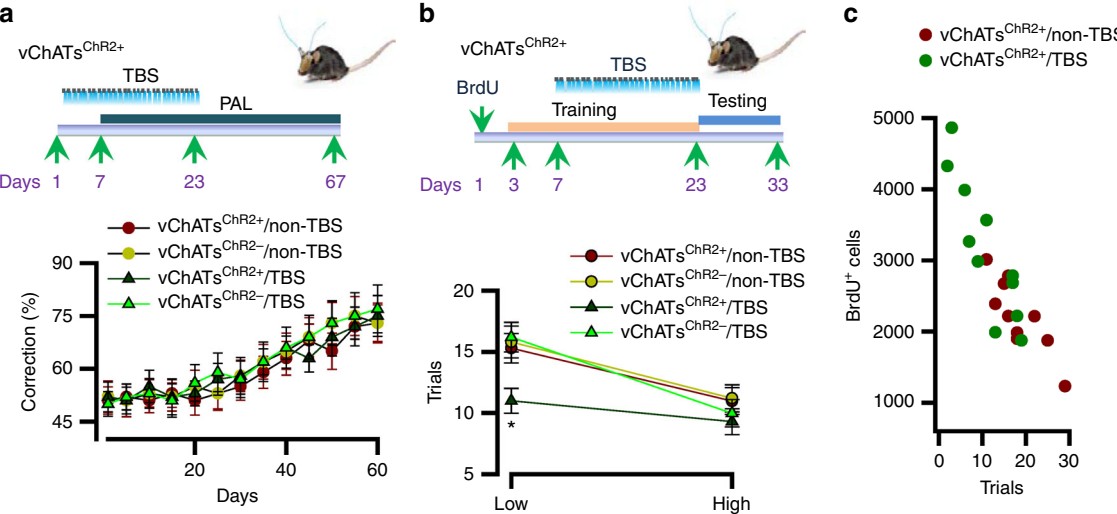

**Fig. 4** Enhancement of spatial pattern separation increases the dNGIs survival. **a** The experimental schedules (top) show that mice at $120 \pm 2$ days old of age were applied with TBS. 7 days after the beginning of TBS, mice were performed with 16-days PAL. The percentage (bottom) of corrections in 60-days PAL is plotted (mean ± SEM, $n = 11$ mice/group). **b** The experimental schedules (top) show that mice (after the completion of PAL) were administered with a single dose of BrdU. 7 days after the administration, mice were trained with 2-CSD. 4 days after the beginning of the training, mice were treated with 16-days TBS. Following the completion of 20-days training, mice were applied for a 5-day block test on each separation (high or low). Immediately after end of the tests, mice were euthanized for BrdU staining. The numbers (bottom) of trials that are required to reach a criterion at a high or a low separation of 2-CSD are plotted (mean ± SEM, $n = 11$ mice/group, $F_{1.43} = 11.02$, *$p = 0.00082$, two-way ANOVA). **c** The numbers of trials that are required to reach a criterion at a low separation in **b** are plotted against the number of BrdU+ cells from the individual mice

the controls (vChATs[NpHR−] mice). These loss-of-function studies demonstrate that the inactivation of vChATs is effective for inhibition of the dNGIs survival.

We next investigated whether a direct synaptic connection between vChATs and dNGIs underlies the dNGIs survival induced by vChATs activation. We blocked cholinergic transmission between vChATs and dNGIs by knocking down the cholinergic M1 receptor gene specifically in dNGIs. We used a recombinant adeno-associated virus (rAAV1/2)-DIO vector that encodes a small interfering RNA (siRNA) specifically targeting the M1 receptor gene (M1siRNA, rAAV1/2-DIO-M1siRNA). A scrambled siRNA (sM1siRNA, rAAV1/2-DIO-sM1siRNA) was used as a control. Virus particles (2 μl), including both the rAAV1/2-DIO-M1siRNA and ΔG-rabies-ChR2 viruses, were injected into the dDG of the dNGIs[TAV/G+/+] mice, leading to the production of dNGIs[M1siRNA+] mice, in which M1siRNA and ChR2 were expressed in dNGIs and vChATs, respectively (Fig. 3g). dNGIs that expressed GFP and M1siRNA (dNGIs[M1-siRNA+] mice) or sM1siRNA (dNGIs[sM1siRNA+] mice) were isolated using cell-sorting strategies immediately after end of the TBS (Fig. 3g). Cell lysates were prepared from the isolated GFP+/mCherry+ dNGIs (yellow, Supplementary Fig. 6a). Our data showed that application of M1siRNA effectively inhibited the M1 receptor mRNA and protein expression of M1 receptor in dNGIs (Fig. 3g and Supplementary Fig. 7). We subsequently performed double whole-cell patch clamp recordings from dNGIs[M1siRNA+] cells paired with the nearby previously existing granule cells (EGCs[M1siRNA−]) in acute brain slices from the dNGIs[M1siRNA+] mice. We demonstrated that M1 receptor currents, as evoked by the illumination of vChATs, were completely blocked in the dNGIs[M1siRNA+] cells; however, they were normal in the dNGIs[sM1siRNA+] cells (Supplementary Fig. 6b). These findings indicate that the expression of M1siRNA effectively inhibits cholinergic transmission in dNGIs.

To determine whether the inhibition of cholinergic transmission in dNGIs impairs the dNGIs survival, we treated the dNGIs[M1siRNA+] and dNGIs[sM1siRNA+] mice without (non-TBS) or with 16-day TBS (TBS). GFP+ cells that were labeled with NeuN

(GFP+) were examined at 28 DAI. Compared with the dNGIs[sM1siRNA+] mice, the dNGIs[M1siRNA+] mice exhibited a substantial reduction in the number of GFP+NeuN+ cells in the dorsal hippocampus ($n = 9$, $F_{1.35} = 7.644$; $P = 0.00093$; two-way ANOVA, Supplementary Fig. 6c). The reduction of GFP+ cells was not a result of the insufficiency of the viral infection because 16-day TBS treatment effectively elevated the numbers of GFP+NeuN+ cells the dNGIs[sM1siRNA+] mice (Supplementary Fig. 6c). Thus, cholinergic transmission in dNGIs plays an essential role in the dNGIs survival in adult mice.

**Cholinergic transmission regulates spatial pattern separation.** To investigate the possibility of a causal relationship between the dNGIs survival induced by the activation of cholinergic transmission and spatial learning and memory, we treated the vChATs[ChR2+] mice without (vChATs[ChR2+]/non-TBS) or with 16-day TBS (vChATs[ChR2+]/TBS). This protocol was selected because it enhanced the survival of dNGIs. In object recognition tests, the vChATs[ChR2+]/TBS and vChATs[ChR2+]/non-TBS mice exhibited comparable levels of exploration of novel and similar objects (Supplementary Fig. 8a–c). In a hidden version of the Morris water maze, both groups spent similar amounts of time searching for a hidden platform in the maze during the training session, as well as similar amounts of time in the target quadrant during the probe trials (Supplementary Fig. 9). Thus, the survival of dNGIs induced by vChATs activation is not associated with recognition memory in object recognition tests and does not involve the task performance in the Morris water maze tests.

We subsequently investigated whether the dNGIs survival induced by vChATs activation influences. A two-choice spatial discrimination task in mice using a touch screen test. This task was selected because it requires spatial pattern separation in the NGC circuits[25]. This apparatus consists of a standard modular chamber equipped with an infrared touch screen, a sugar pill dispenser, an illuminated receptacle with head-entry detectors, and a tone generator. Mice were pre-trained to learn an association between a tone and a reward and an association

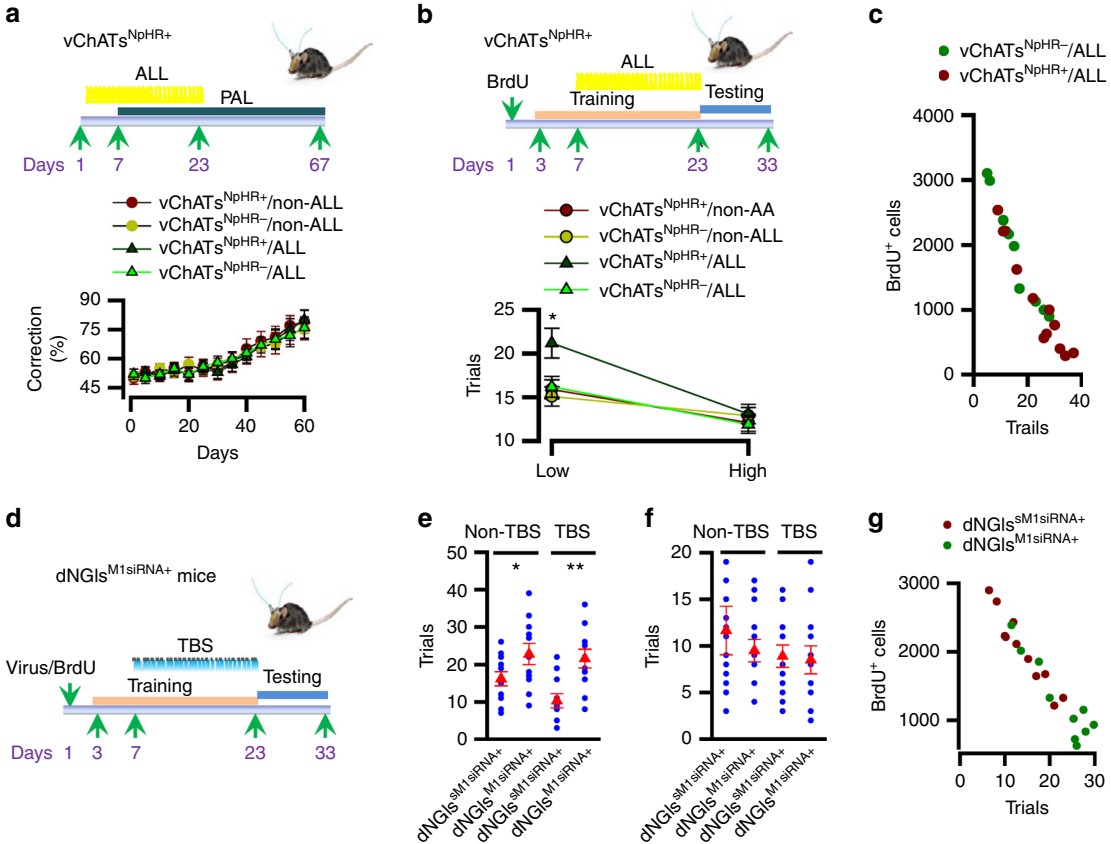

**Fig. 5** Inhibition of spatial pattern separation reduces the dNGIs survival. **a** The experimental schedules (top) show that mice at $120 \pm 2$ days old of age were applied with ALL. 7 days after the beginning of ALL, mice were performed with 16-days PAL. The percentage (bottom) of corrections in 60-days PAL is plotted (mean ± SEM, $n = 11$ mice/group). **b** The experimental schedules (top) show that mice (after the completion of PAL) were administered with a single dose of BrdU. 7 days after the administration, mice were trained with 2-CSD. 4 days after the beginning of the training, mice were treated with 16-days ALL. The mice were applied for a 5-day block test on each separation (high or low). Immediately after end of the tests, mice were euthanized for BrdU staining. The numbers (bottom) of trials that are required to reach a criterion at a high or a low separation of 2-CSD (mean ± SEM, $n = 11$ mice/group, $F_{1.43} = 9.87$; *$p = 0.00098$, two-way ANOVA). **c**, The numbers of trials that are required to reach a criterion at a low separation in **b** are plotted against the number of BrdU+ cells from the individual mice (Rsqr = 0.92 in vChATs$^{NpHR-}$/ALL vs. 0.94 in vChATs$^{NpHR+}$/ALL mice). **d–f** The experimental schedules **d**, and the actual (blue circles) and averaged trials that are required to reach a criterion at a low **e**, or a high **f**, separation of 2-CSD from the individual mice (mean ± SEM, $n = 11$ mice/group, $F_{1.43} = 9.79$, *$p = 0.0012$, **$p = 0.00082$, two-way ANOVA). **g** The numbers of trials that are required to reach a criterion at a low separation in **e** are plotted against the number of BrdU+ cells from the individual animals that were treated with TBS (Rsqr = 0.936 in dNGIs$^{sM1siRNA+}$ vs. 0.648 in dNGIs$^{M1siRNA+}$ mice)

between a response at the screen and earning a sugar pill reward. Following the successful completion of the pre-training, mice were trained on a touch-screen for paired-association learning (PAL), in which mice were required to learn the paired association of three objects (flower, plane and spider) with their correct spatial locations (left, middle and right). Mice were rewarded when they identified a correct pair of object and position (a correct object in its correct position) during a choice between two objects; one object was paired with a correct location, whereas the other object was paired with one of the two incorrect locations. Our data showed that both vChATs$^{ChR2+}$ and vChATs$^{ChR2-}$ mice treated without (non-TBS) and with TBS had the similar accuracy rates in PAL (mean ± SEM; $n = 11$ mice/group, Fig. 4a).

We next examined the performance in a two-choice spatial discrimination (2-CSD) task. Animals were required to identify the correct spatial location between two illuminated boxes that were separated by 1 (low separation) or 3 (high separation) empty positions. An analysis of the choice accuracy, as defined by earning the rewards, indicated that vChATs$^{ChR2+}$ mice that were treated with TBS performed significantly better than those treated

without TBS at a low separation task (Fig. 4b). A better performance in a 2-CSD task is closely associated with an increase of BrdU labeling (Fig. 4c).

We subsequently determined whether spatial pattern separation relies on the dNGIs survival. We inactivated vChATs by delivering ALL directly onto vChATs for 16 consecutive days in vChATs$^{NpHR+}$ mice (Fig. 5a). Delivery of ALL effectively inhibited spatial pattern separation (Fig. 5b) and this inhibition correlated with a reduction of the BrdU labeling (Fig. 5c). Next, we blocked cholinergic synaptic transmission by silencing M1 receptor gene in dNGIs (Fig. 5d). We found that inhibition of cholinergic synaptic transmission in dNGIs completely blocked the beneficial effects of TBS in both spatial pattern separation (Fig. 5e) and the BrdU labeling (Fig. 5f, g and Supplementary Fig. 6c). Thus, cholinergic transmission in dNGIs regulates spatial pattern separation via the dNGIs survival.

**Cholinergic transmission is degenerated in AD mice.** Loss of spatial memory is the earliest clinical sign of AD[6, 28], in which both synaptic transmission and neurogenesis in the hippocampus

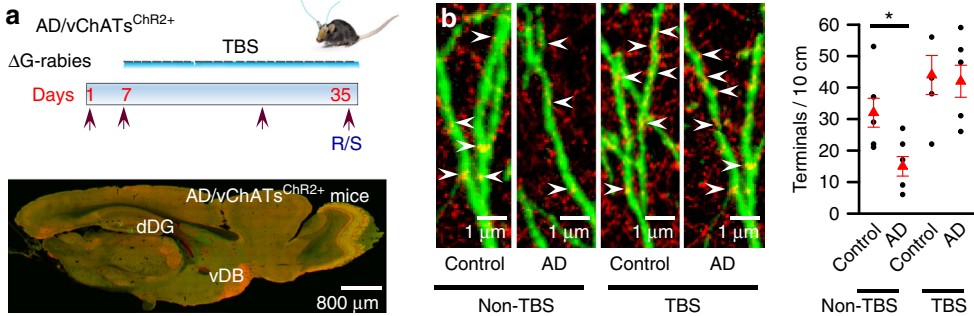

**Fig. 6** TBS increases cholinergic synaptic inputs in AD mice. **a** Experimental schedules (top) show that ΔG-rabies-ChR2 virus injection in the AD/NGIs^TVA/G+ mice at 150 ± 5 days old of age resulted in AD/vChATs^ChR2+ mice. The mice were applied with 28-days TBS at 7 days after the virus injection. Electrophysiological recordings and staining (R/S) were performed immediately after TBS. A representative image indicates the ChR2-GFP expression in vChATs neurons of the AD/vChATs^ChR2+ mice. **b** Representative images reveal the ChATs-labeled terminals targeting at the individual dNGIs. A plot shows the actual (black circles) and the averaged numbers (red triangles) of terminals/10 cm dendrites (mean ± SEM, $n = 5$ mice/group, $F_{1.19} = 10.8$, *$p = 0.0051$, two-way ANOVA)

are impaired[7, 29]. Thus, we hypothesized that in AD mice, cholinergic transmission between vChATs and dNGIs is impaired and this impairment defects the pattern separation-associated spatial memory. To test this hypothesis, we examined cholinergic synaptic terminals that directly innervate dNGIs in AD mice (homozygous Tg2576-APPswe mice in a C57BL/6 genetic background), which have Aβ plaque pathologies similar to AD patients[26]. We crossed AD mice with dNGIs^GFP+/+ mice in AD/dNGIs^GFP+/+ mice, in which GFP was expressed restrictively in dNGIs when treated with TAM. Brain sections from the AD/dNGIs^GFP+/+ mice and the age-matched non-AD controls (control/dNGIs^GFP+/+ mice) were stained with an antibody against ChAT protein (Supplementary Fig. 10a). The densities of the ChAT-labeled terminals that were associated with dendrites of dNGIs were 48.9% lower in AD/dNGIs^GFP+/+ mice, compared with control/dNGIs^GFP+/+ mice (18 ± 2.1 terminals/10 cm vs. 35 ± 3.1 terminals/10 cm dendritic branches, respectively, mean ± SEM, $n = 32$ cells/5 mice/group, $p < 0.01$; two-way ANOVA with Bonferroni post-hoc test, Supplementary Fig. 10b). Thus, cholinergic synaptic inputs onto dNGIs in AD mice are reduced.

We next determined whether a reduction of the ChAT-labeled terminals is associated with the defects of cholinergic synaptic transmission between vChATs and dNGIs. We engineered vChATs in AD mice with the expression of channelrhodopsin-2-E123A (ChR2). To this end, we created a mutant strain of mice with an inducible expression of the avian viral receptor TVA and rabies G-GFP in dNGIs (dNGIs^TVA/G+/+ mice) by crossing TVA/G^loxP/loxP mice with Nestin-Cre^ER+/+ mice. The dNGIs^TVA/G+/+ mice were crossed with AD mice. Following oral administration of TAM (100 mg in corn oil/kg body weight), TVA/G-GFP were expressed in dNGIs of AD mice (AD/dNGIs^TVA/G+/+ mice). Next, we created ΔG-rabies viruses that encoded DIO-ChR2 (ΔG-rabies-DIO-ChR2 virus). A high titer (1 μl, 2 × 10^11 genomic particles/ml) of virus particles was injected into the dDG of AD/dNGIs^TVA/G+/+ mice, in which ChR2 was expressed in dNGIs and their direct presynaptic vChATs (AD/vChATs^ChR2+ mice, Fig. 6a).

We next recorded cholinergic synaptic transmission between vChATs and dNGIs. Brain slices were prepared from AD/vChAT^ChR2+ mice and age-matched control/vChATs^ChR2+ mice at 16 days after the injection (DAI) of ΔG-rabies-DIO-ChR2 virus particles. Whole-cell patch clamp recordings of synaptic currents in dNGIs were evoked by illumination of vChATs^ChR2+ cells with increasing intensities (1–5 mV). The normalized peak amplitude (NPA) of the currents; the peak currents that were evoked at an intensity of 5 mV/mm² were normalized to those

evoked at an intensity of 1.0 mV/mm² (defined as 1.0) was comparable between AD/vChATs^ChR2+ mice at 150 ± 2 days of age and the age-matched control/ChATs^ChR2+ mice (69.6 ± 5.3 vs. 64.3 ± 6.2, respectively, mean ± SEM, $n = 39$ cells/9 mice/group; $p > 0.05$, t-tests; Supplementary Fig. 10c); however, it was significantly decreased in AD/vChATs^ChR2+ mice at 180 ± 5 days of age, compared with the age-matched control/vChATs^ChR2+ mice (23.8 ± ± 2.6 vs. 69.3 ± 6.7, respectively, mean ± SEM, $n = 39$ cells/9 mice/group; *$p < 0.01$, two-way ANOVA with Bonferroni post-hoc test, Supplementary Fig. 10d). Together, these findings indicate that cholinergic synaptic transmission is impaired in AD mice.

**Spatial learning and memory is impaired in AD mice**. To determine whether the failure of cholinergic synaptic transmission in dNGIs impairs spatial memory in AD/vChATs^ChR2+ mice, we examined the behavioral performance of AD/vChATs^ChR2+ mice paired with control mice. In the Morris water maze tests, although the capability for spatial information acquisition was highly variable from mouse to mouse, on average, AD/vChATs^ChR2+ mice at 180 ± 5 days of age spent more time searching for a hidden platform in the maze during the training sessions and less time in a targeting quadrant during the probe trials than the age-matched control/vChATs^ChR2+ mice (Supplementary Fig. 11a, b). In the open field, AD/vChATs^ChR2+ mice at 180 ± 5 days of age exhibit motor activities similar to those of their age-matched controls. These data are consistent with most previous reports that spatial learning and memory but not motor activity is impaired in AD mice at 180 days of age[28, 29].

To further determine the deficits of spatial learning and memory in AD mice, we examined spatial pattern separation. AD and the age-matched control mice had the similar accuracy rates in PAL even when they were at 180 days old of age (Supplementary Fig. 11c), whereas control mice at 180 days old of age performed significantly better than the age-matched AD mice at a low separation task of 2-CSD (Supplementary Fig. 11d). Together, these data reveal that spatial learning and memory is impaired in AD mice at 180 days old of age.

**TBS improves spatial pattern separation in AD mice**. Deep brain stimulation (DBS) is therapeutically effective against the disease progression in several neurological disorders, including AD and Parkinson's disease[26, 30, 31]. We therefore investigated whether DBS is able to protect against the impairments of cholinergic synaptic transmission and improve the neurological

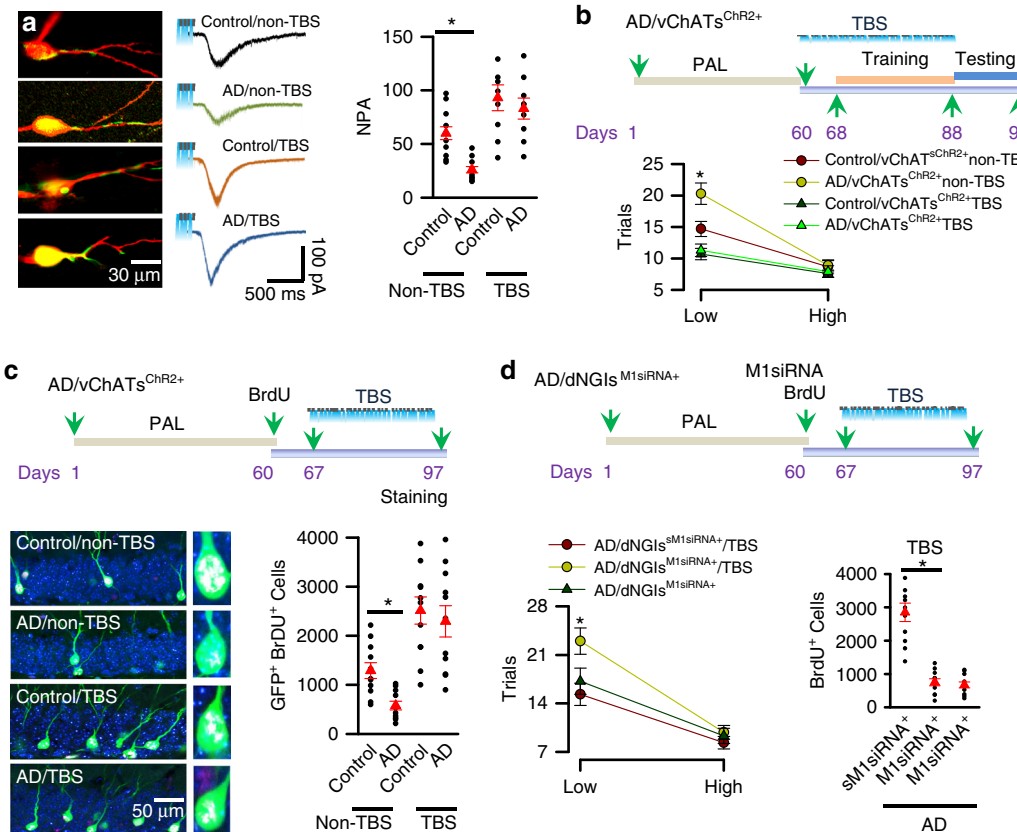

**Fig. 7** TBS improves spatial pattern separation in AD mice. **a** Representative images (left) and recording (middle) from the individual dNGIs (green) filled with Lucifer yellow (red) through the recording electrodes in slices (left) from AD/vChATs$^{ChR2+}$ and control/vChATs$^{ChR2+}$ mice treated without (non-TBS) or with TBS. The cholinergic synaptic currents were evoked by illuminating vChATs at an intensity of 5 mV. A plot (right) shows the actual (black circles) and averaged values (red triangles) of the normalized peak amplitude (NPA) of the currents at an intensity of 5 mV. The peak currents were normalized to those evoked by an intensity of 1 mV (defined as 1.0, mean ± SEM, $n = 9$ mice/group, $F_{1.35} = 10.91$, *$p = 0.0019$, two-way ANOVA). **b** Experimental schedules show that AD/vChATs$^{ChR2+}$ mice at 80 ± 2 days old of age were performed with PAL. After the completion of PAL, the mice were treated with 28-days TBS. A plot (bottom) shows the trials required to reach a criterion in AD/ChATs$^{ChR2+}$ and control/vChATs$^{ChR2+}$ mice treated without (non-TBS) or with TBS at a low or a high separation (mean ± SEM, $n = 11$ mice/group, $F_{1.43} = 7.99$, *$p = 0.00072$, two-way ANOVA). **c** Experimental schedules and representative images and the actual (black circles) and averaged numbers (red triangles) of GFP$^+$BrdU$^+$ cells in the individual AD/ChATs$^{ChR2+}$ and control/vChATs$^{ChR2+}$ mice treated without (non-TBS) or with TBS (mean ± SEM, $F_{1.35} = 38.93$, *$p = 0.00081$, $n = 9$ mice/group, two-way ANOVA). **d** Experimental schedules (top) show that AD/dNGIs$^{M1siRNA+}$ mice at 80 ± 2 days old of age were performed with PAL. After the completion of PAL, mice were administered with a single dose of BrdU and M1siRNA virus. BrdU staining was done immediately after TBS. The trials (bottom) required to reach the criterion at a low or a high separation (mean ± SEM, $n = 11$ mice/group, $F_{1.32} = 35.07$, *$p = 0.00097$, two-way ANOVA). The actual (black circles) and averaged numbers (red triangles) of BrdU$^+$ cells in the individual mice are plotted (mean ± SEM, $n = 11$ mice/group, $F_{1.32} = 68.61$, *$p = 0.00026$, two-way ANOVA)

function of AD mice. Hippocampal theta oscillations encode an animal's position during spatial navigation[26]. Thus, we applied a 28-day theta bust stimulation (TBS) paradigm in AD/vChATs$^{ChR2+}$ mice beginning at 150 ± 5 days of age (Fig. 6a). Four groups of mice including AD/vChATs$^{ChR2+}$ and age-matched control/ChATs$^{ChR2+}$ mice treated with (TBS) or without (non-TBS) using TBS. TBS was delivered using 0.2-mm optical fibers bilaterally onto the vChATs once per day for 28 consecutive days. Immediately after the TBS treatment, brain sections were stained with anti-ChAT antibody. The densities of cholinergic synaptic terminals that innervate the dendrites of dNGIs in AD/vChATs$^{ChR2+}$ that were treated with TBS were identical to control/vChATs$^{ChR2+}$ mice (Fig. 6b). Consistent with these data, synaptic currents evoked by illumination of vChATs became comparable between AD/vChATs$^{ChR2+}$ mice at 180 ± 5 days of age and age-matched control/vChATs$^{ChR2+}$ mice (Fig. 7a).

We next determined the protective effects of TBS in spatial learning and memory. TBS did not alter the performance in a hidden version of the Morris water maze (Supplementary

Fig. 12a–c) and had no effect on the motor activity, novel object recognition, body weight (Supplementary Fig. 12d–g) and a PAL task (Supplementary Fig. 13), however, it significantly improved the performance of AD/vChATs$^{ChR2+}$ mice at a low separation of 2-CSD (Fig. 7b) and increased the numbers of BrdU$^+$ cells (Fig. 7c).

To determine whether the improvements of spatial pattern separation in AD/vChATs$^{ChR2+}$ mice that were treated with TBS are due to activation of cholinergic synaptic transmission and the dNGIs survival, we tested whether inhibition of cholinergic synaptic transmission in dNGIs eliminates the protective effects of TBS in AD mice. We knocked down cholinergic M1 receptor gene specifically in dNGIs using rAAV1/2-DIO-M1siRNA virus vector. Virus particles (2 μl), including both rAAV1/2-DIO-M1siRNA and ΔG-rabies-ChR2 viruses, were injected into the dDG of AD/dNGIs$^{TAV/G+/+}$ mice, resulting in the production of AD/dNGIs$^{M1siRNA+}$ mice, in which M1siRNA and ChR2 were expressed in dNGIs and vChATs respectively. Knocking down M1 receptor gene completely eliminated the TBS-induced

enhancement of spatial pattern separation and BrdU labeling (Fig. 7d). Thus, cholinergic synaptic degeneration contributes to the impairments of spatial pattern separation via the inhibition of the dNGIs survival in AD mice.

The roles of cholinergic synaptic degeneration in the spatial memory loss were further investigated by using an additional mouse model of AD, in which both the APPKM670/671NL (Swedish) and the PSEN1deltaE9 mutant geneses were expressed (AD1 mice). We crossed AD1 mice with the vChATs[ChR2+] and resulted in the AD1/vChATs[ChR2+] mice (Supplementary Fig 14a). In a low separation of 2-CSD task, the performance of AD1/vChATs[ChR2+] mice was significantly reduced, compared to the age-matched control/vChATs[ChR2+] mice (Supplementary Fig. 14b, c) and this reduction of the performance was associated with a decrease of BrdU labeling in the dDG region (Supplementary Fig. 14d). When AD1/vChATs[ChR2+] mice were treated with TBS, pattern-separation-associated spatial memory was significantly improved (Supplementary Fig. 14e). These data using an additional AD mouse model confirm that the capacity of the dNGIs survival is reduced and this reduction of the dNGIs survival contributes to the impairments of spatial pattern separation in AD mice.

## Discussion
Our present study explored a cholinergic synaptic mechanism underlying memory loss in AD, with the following three key findings. First, we uncovered that vChATs establish functional synaptic connections with dNGIs and regulate pattern separation-associated spatial memory. Second, we demonstrated that cholinergic synaptic transmission is impaired and this impairment is associated with the loss of spatial pattern separation in the early stage of AD. Third, we unraveled that activation of vChATs via a TBS paradigm protects against the impairments of spatial pattern separation by intervention of cholinergic synaptic impairments in dNGIs of AD mice. Together, our present study not only reveals a previously unknown cholinergic pathway that modulates pattern separation-associated spatial memory but also provides potential target for therapeutic interventions of the spatial memory loss in AD.

Previous studies reported that lesion of cholinergic nuclei or transection of the fimbria-fornix cholinergic fibers impairs learning and memory[17, 32, 33]. Elevation of the ACh levels by pharmacological inhibition of acetylcholinesterase improves behavioral performance in AD[34]. However, the experimental approaches used in these studies generate tissue damages and possibly involved the deletion of non-cholinergic pathways and/or non-specific activation of cholinergic synapses. Direct evidence for cholinergic regulation of spatial memory remains lacking. In our present studies, we established a monosynaptic tracing and recording strategies that selectively target cholinergic transmission in AD mice. Using these strategies, we identified that cholinergic synaptic transmission in dNGIs plays an essential role in spatial pattern separation.

Our present studies demonstrate that M1 receptor mediates cholinergic transmission in dNGIs of adult mice. Previous studies have indicated that several subtypes of cholinergic receptors are expressed in neuronal cells throughout the hippocampus[35], and several of these receptor types, including α7- and β2-containing acetylcholine receptors (α7nAchR and β2nAchR), are associated with the proliferation or survival of NGIs[36]. However, these previous studies depended on conventional knockout strategies, in which the α7nAchR or β2nAchR gene was deleted in all neuronal cells throughout the mouse brain. This strategy generates non-cell-autonomous actions of cholinergic receptors in NGIs and represents receptor-dependent network activity that

impinges on previously existing neuronal cells. In our present studies, we specifically targeted M1 receptor gene in dGNIs only and unraveled that activation of M1 receptor is an essential for cholinergic synaptic transmission in dNGIs. Consistent with this conclusion, several previous studies indicate that M1 receptor mediates cholinergic transmission in the adult-born neurons[37, 38].

DBS shows promise as a therapeutic approach for some neurological disorders[39], including AD[30, 40], Parkinson's disease[41], epilepsy[42] and depression[43]. A recent work reveals that stimulation of cortical cells with 40 Hz for only once a day is able to generate a long-lasting inhibition of the Aβ peptide accumulation in AD mice[44]. Activation of a small group of neural cells in the hippocampus was also found to increase the numbers of dendritic spines and improve the spatial learning and memory in AD mice[45]. Together with our findings[26, 46], it is plausible that a brief activation of a small population of neurons in the brain can restore cellular health and circuit integrity possibly via the release of transmitters and the growth factors that are required for neuronal cell survival and regeneration. Consistent with this, we have recently found that a brief activation of cortical neurons with TBS is able to inactivate cell-death associated protein kinase 1(DAPK1) and hence protects against the impairments of excitatory synaptic transmission in AD mice[46].

Our present studies show that the impairments of the dNGIs survival in the dorsal hippocampus contribute to the decays of pattern-separation-associated spatial memory in AD mice. It has been known that the hippocampal neurogenesis in adults can be modified by some neurological disorders such as stress[47, 48], depression[49] and schizophrenia[50] as well as aging[51]. Specifically, the basal levels of the dentate neurogenesis are very low in older animals including both mice and rats, compared to young adults[49–52]. Significantly, a decrease in the number of new neurons in the hippocampus was found to be closely associated with the aging-related declines of cognition such as learning and memory[52–54]. While a reduction of the hippocampal neurogenesis was reported in normal mice and rats only when they were at 8 months old of age or older[51–54], the declines of the dNGIs survival in AD mice were seen even when they were at 6 months old of age, suggesting that the mechanisms underlying the defects of the dNGIs survival in AD mice differ from these seen in normal aging mice. Indeed, in AD mice a decrease of the new neuron survival was found in the dorsal but not in the ventral portion of the hippocampus. Facilitation of the dNGIs survival in AD mice improves spatial pattern separation, but not alters spatial navigation in the Morris water maze. Thus, further studies are required to define how the survival of adult-born cells in the ventral portion of the hippocampus is regulated and whether the survival of the ventral hippocampal cells contributes to pattern separation-independent spatial memory.

## Methods
**Animals**. Male mice at $120 \pm 2$ days old of age were used, unless otherwise specified. Mice were bred and reared under the same conditions in accordance with institutional guidelines and the Animal Care and Use Committee (Huazhong University of Science and Technology, Wuhan, China) of the university's animal core facility (UAC) and housed in groups of three to five mice/cage under a 12-h light-dark cycle, with lights on at 8:00 am, at a consistent ambient temperature ($21 \pm 1\,°C$) and humidity ($50 \pm 5\%$). All experiments and analyses were performed blind to the genotype or treatment. AD (Tg2576-APPswe) mice, ChAT-Cre knock-in mice (stock No: 006410) and Nestin-Cre[ER] mice (stock No: 016261) were purchased from Jackson Laboratory (Bar Harbor, ME, USA) and were identified as homozygous and bred at the UAC. The strains of the mutant mice used in this study are listed in the Supplementary Table 1 and the methods for breeding are described in the figure legend.

**Monosynaptic tracing in adult mice in vivo**. To determine a direct synaptic connection between vChATs and dNGIs, we created a mutant line of mice with a *loxP*-flanked STOP sequence followed by the avian viral receptor TVA and rabies

G with or without GFP (TVA/G$^{loxP/loxP}$ mice)[26, 46]. The Rosa-CAG-Flag-TVA/G-WPRE targeting vector was designed with a CMV-IE enhancer/chicken beta-actin/rabbit beta-globin hybrid promoter (CAG), an FRT site, a loxP-flanked STOP cassette, a Flag-TVA/G (or TVA/G-GFP) sequence, a woodchuck hepatitis virus post-transcriptional regulatory element (WPRE; to enhance the mRNA transcript stability), a poly-A signal, and an attB/att-flanked PGK-FRT-Neo-poly-A cassette. This entire construct was inserted into the Gt(ROSA)26Sor locus via electroporation of C57BL/6-derived embryonic stem (ES) cells. The targeted ES cells were injected into C57BL/6 blastocysts. The chimeric mice were bred to C57BL/6 mice. TVA/G$^{loxP/loxP}$ mice were crossed with Nestin-Cre$^{ER+/+}$ mice, leading to the production of the Nestin$^{TAV/G+/+}$ mice, in which TVA/G (or TVA/G-GFP) expression in the proliferating cells in the adult hippocampus was induced by the administration of TAM (100 mg in corn oil/kg body weight/day for three consecutive days). The administration of corn oil alone was used as a vehicle control. A high titer (2 µl of $7 \times 10^{10}$ genomic particles/ml) of the ΔG-rabies virus that encoded mCherry was stereotaxically injected into the dorsal dentate gyrus (dDG) of Nestin$^{TVA/G+/+}$ mice, which caused a specific labeling of dNGI presynaptic neurons, including pyramidal neurons in the entorhinal cortex and vChATs in the vDB.

**Virus vectors and injections**. We used the recombinant adeno-associated virus (rAAV1/2) that expressed CAG-driven M1siRNA (GCCATCCTCTTCTGGCAAT)[26, 46, 55, 56]. The control vector expressed sM1siRNA (TTCTCCGAACGTGT-CACGT). We designed the rAVE-M1siRNA or sM1siRNA vector via the insertion immediately downstream of the CAG translational STOP codon through *Apal/Kpnl*. The rAVE plasmids were co-transfected with the AAV helper1/2 mixers into HEK293 cells to generate the rAAV1/2-M1siRNA or sM1siRNA virus particles with a high titer ($>5 \times 10^{12}$ genomic particles/ml). Virus particles (1 µl) were bilaterally injected into the dDG (ap1.7/ml1.0/dv2.1).

For monosynaptic retrograde tracing of dNGI pre-synaptic neurons (vChATs), a high titer (2 µl of $3 \times 10^9$ genomic particles/ml, provided by Dr. Fuqiang Xu at the Wuhan Institute of Physics and Mathematics, Chinese Academy of Sciences) ΔG-rabies virus that encoded mCherry (ΔG-rabies-mCherry, ΔG-rabies-ChR2-IRES-mCherry or ΔG-rabies-NpHR-IRES-mCherry) was stereotaxically injected into the dDG in the Nestin$^{TVA/G+/+}$ mice. This approach produced a specific monosynaptic retrograde labeling of vChATs in the vDB.

**Optogenetics in vitro**. To investigate functional cholinergic transmission between vChATs and dNGIs, we expressed ChR2 or NpHR in vChATs via the injection of a high titer ($3 \times 10^{10}$ genomic particles/ml) of the ΔG-rabies virus that encoded ChR2 or NpHR into the dDG of the Nestin$^{TVA/G+/+}$ mice. 3 days after the injection, ChR2 (the ΔG-rabies-DIO-ChR2 virus, ChATs$^{ChR2+}$ mice) or NpHR (the ΔG-rabies-DIO-NpHR virus, ChATs$^{NpHR+}$ mice) was expressed in the dNGIs pre-synaptic ChATs. We then prepared hippocampal slices (300 µm) from the ChATs$^{ChR2+}$ mice or ChATs$^{NpHR+}$ mice, as described before[26, 46, 55, 57]. The slices were transferred to a holding chamber that contained artificial cerebrospinal fluid (ACSF, in mM: 124 NaCl, 3 KCl, 26 NaHCO₃, 1.2 MgCl₂•6H₂O, 1.25 NaH₂-PO₄•2H₂O, 10 C₆H₁₂O₆, and 2 CaCl₂ at pH 7.4, 305 mOsm) at 32 °C for 30 min. The temperature was then regulated at 22 °C for 60 min. A slice was selected and transferred to a recording chamber, which was continuously perfused with oxygenated ACSF (2 ml/min) at 22 °C. We performed whole-cell patch clamp recordings from GFP⁺ dNGIs in a slice, which was visualized under a fluorescent infrared-phase-contrast (IR-DIC) Axioskop 2FS upright microscopy, which was equipped with a Hamamatsu C2400-07E infrared camera. Synaptic currents were evoked using a 473-nm laser (DPSS laser, Anilab). The laser power was ranged from 0.1 to 5 mW/mm². The internal solution in the recording electrodes consisted of (in mM) 140 potassium gluconate, 10 HEPES, 0.2 EGTA, 2 NaCl, 2 MgATP, and 0.3 NaGTP. The external ACSF solution contained glutamate and GABA$_A$ receptor antagonists including 10 µM bicuculline (Cat. No. 0130; TOCRIS), 30 µM CNQX (Cat. No. 0190; TOCRIS) and 50 µM APV (Cat. No. 0106; TOCRIS) and nicotinic acetylcholine receptor antagonist (Ni, tubocurine, Cat. No. 2820) or M1 receptor antagonist VU0255035 (Cat. No. 3727; TOCRIS) or M2 receptor antagonist AFDX116 (Cat. No. 1106; TOCRIS) or M3 receptor antagonist DAU5884 (Cat. No. 2096; TOCRIS) or M4 receptor antagonist PD102807 (Cat. No. 1671; TOCRIS). The row electrophysiological data were collected at 10 kHz and filtered with a low-pass filter at 2 kHz and was analyzed using ClampFit 10.2 software (Molecular Devices) with template matching at a threshold of 5 pA.

**Optogenetics in vivo**. We anesthetized mice with 6% chloral hydrate (0.06 ml/10 g; intraperitoneally) and coated four tetrodes of twisted 17-µm HM-L with platinum-iridium (10 or 20% platinum, #: 100–167, California Fine Wire Company) and connected tetrodes to a microdrive for dorsal-ventral adjustment after the implantation. We positioned the tetrodes directly above the recording site and secured the micro-drive to the skull using jeweler's screws and dental cement. A jeweler's screw was used as a ground electrode. We screened cells and behaviors daily for each experimental procedure. During the screening procedures, we lowered the tetrodes slowly over several days in steps of 30 µm. For light activation of ChATs in the vDB region, we bound a 200-µm-diameter, unjacketed optical fiber (Ocean Optics) to a tetrode-containing silicone tube (166 µm) and bilaterally

implanted the fibers into the vDB (ap0.9/ml0.4/dv5.2). The fiber/tetrode complex was secured to the skull using jeweler's screws and dental cement. We validated position of optic fibers by electrolytic lesions after light stimulation. We applied a 473-nm laser (DPSS laser, Anilab) for generation of light pulses and TBS. The laser power that was used to activate cholinergic synaptic transmission was ranged from 0.1 to 5 mW/mm². A TBS paradigm consisted of 10 trains of stimuli at 10 s intervals. Each train consisted of 4 spikes at 100 Hz. TBS was applied for 16 consecutive days (16-days TBS) in normal adult mice and for 28 consecutive days (28-days TBS) in AD mice.

**BrdU labeling and immunohistochemistry**. We administered mice with a single dose of BrdU (230 mg/kg body weight, i.p.). 1, 3, 7, 11, 14, 21, or 28 days after the BrdU injection (DAI), mice were processed for the experiments. Specifically, the mice were anesthetized with chloral hydrate (30 mg/kg body weight) and transcardially perfused with ice-cold PBS followed by ice-cold 4% PFA in PBS. The brains were post-fixed overnight in 4% PFA in PBS. Free-floating sections (~ 30 µm thick, one-sixth of the total sections that contained the dentate gyrus) were sliced. For BrdU staining, the sections were heated (85 °C for 5 min) in antigen-unmasking solution (Vector Laboratories, Burlingame, CA), incubated in 2 M HCl (37 °C for 30 min) and then in 0.1 M boric acid/NaOH (pH 8.5, room temperature for 15 min) and blocked in 3% BSA (room temperature for 1 h). After the blocking, the sections were incubated with rat monoclonal anti-BrdU (1:300; AbD Serotec) in 0.1 M PBS with 3% BSA and 0.5% Triton X-100 for overnight. The sections were then washed and developed using an ABC Kit (1:400; Vector). For fluorescence labeling, we incubated the brain sections with anti-ChAT (1:1000 Abcam) or anti-NeuN (1:500 Millipore), or GFAP (1:1000 Millipore) or anti-DCX (1:1000 Abcam) antibodies for overnight. After the incubation with the primary antibodies, the sections were washed and incubated with Alexa 488-, 546- or 647-conjugated secondary antibodies (1:500; Invitrogen) for 45 min. In some experiments, some slices were also incubated with DAPI (1:1000; Sigma) for 10 min after incubation with the secondary antibodies. An inverted laser-scanning confocal microscope (LSM 780; Zeiss) was used for fluorescence imaging. All quantifications were performed by investigators, who blinded to the experimental conditions.

**Open-field and object recognition tests**. We measured motor activity within clear boxes that measured $100 \times 100$ cm² and outfitted with photo-beam detectors for monitoring horizontal and vertical activity[26, 56, 57]. The data were analyzed using MED Associates Activity Monitor Data Analysis software. The mice were placed in a corner of the open-field apparatus and allowed to move freely. The variables recorded included the resting time (s), ambulatory time (s), vertical/rearing time (s), jump time (s), stereotypic time (s) and average velocity (cm/s). The mice were not exposed to the chamber prior to testing. The data were recorded for each individual animal during 30-min intervals.

To test the performance in the object recognition task, we subjected mice for two sessions of one trial each: acquisition and retrieval trials. During the acquisition trial, we placed mice in an arena that contained two identical objects for 5 min. The mice that did not explore the objects for 20 s in the 5 min period were excluded from the further experiments. We defined exploration as a mouse approaching its nose within 1 cm of the object. This approaching was associated looking, sniffing, or touching. The retrieval session was done 2 h after the acquisition trial. In this trial, we replaced one of the objects presented in the first trial with a novel object. We then placed mice back in the arena for 5 min and recorded the total time spent in the exploration of each object. Motor activity and time spent in active exploration of the familiar (F) or novel (N) objects during the retrieval trial were calculated. Recognition memory was evaluated using a recognition index (RI) for each mouse with a formula $((N-F)/(N + F)) \times 100$. RI reflected the difference between the time exploring the novel and familiar objects and the total time exploring both objects.

**Morris water maze**. We filled a 1.5 m-diameter swimming pool with white and non-toxic ink water. Pool temperature was maintained at 25 °C. We brought mice to the behavior room where they were housed for the training for 1–2 days before training session[26, 46, 57]. The training session lasted for 7 days. In the first day of training, mice were allowed to rest on the platform for 30 s and to have 60 s for finding the hidden platform. In case that a mouse did not find the platform within 60 s, we guided this mouse to find and stay the platform for 30 s. Throughout the period of training session, mice were required for perform a total of 4 trials, in which mice were released at four different randomized release points of the pool. Immediately, after the 6-day training session, mice were required to perform a one-probe trial. In both training and probe trials, the behavioral tests were performed by an experimenter who was unaware of the genotypes and treatments.

**Pattern separation**. We trained mice in a Bussey-Saksida Touch Screen chamber system (ABET II, Lafayette Instrument Company, IN, USA), which had black plastic walls and a perforated stainless steel floor. One end of the box was fitted with a touch-sensitive flat screen equipped with infrared photocells to detect a mouse nose touching. The other end of the box contained a food dispenser with switchable illumination plus an infrared beam to detect mouse entries. The boxes were also equipped with a fan, a tone and click generator, a sugar pill food

dispenser and LED light for general illumination. The behavioral performance was monitored using an infrared digital camera with Whisker software and an ABET II system. Percentage accuracy = (100 × (correct responses)/(correct + incorrect responses)). Prior to paired-associate learning (PAL), mice were housed under a 12-h light cycle (lights off at 0700 hours) for 7 days. After this period of time, food was restricted to maintain 85–90% of free feeding body weight. Mice were then pre-trained to touch the screen for a sugar pill reward and to break the infrared beam near the sugar pill tray for an initiation of trials. Reward collection for a correct nose touch was followed by a 15-s intertribal interval. An incorrect response resulted in a time-out (5 s) and the presentation of a correction trial until a correct response was made. The purpose of correction is to counteract side and stimulus biases and to ensure that all mice receive a consistent number of rewards per session despite their performance on incorrect trials. Mice were moved onto PAL tests after the completion of 48 trials within 30 min at over 80% corrects for two consecutive sessions.

In a PAL task, mice at 90 ± 1 days old of age were trained to learn one of three visual images (spider, plane, or flower). Two images were displayed on a touch screen. One image was always paired with a correct location while the other was paired with an incorrect location. In case that a mouse selected the correct image/location pair, a reward was delivered; If the incorrect image per location pair was selected, no reward was administered.

Following the completion of PAL, the two-choice spatial discrimination (CSD) tests were performed. During an initial 20-day training period, we trained mice to touch one of two illuminated squares (e.g., the left-most square) until a criterion of consecutive touches was reached. When the criterion was reached, the other location (e.g., now the right-most square) was designated correct. Mice were allowed only 1 reversal a day with a maximum of 41 trials a day and a criterion of 9 of 10 consecutive touches. The separation of the illuminated boxes on the touch screen was presented with a high degree of separation (i.e., separation 4 = 3 empty per dark locations between the two illuminated locations) or a low degree of separation (i.e., separation 2 = 1 empty per dark location between the two illuminated locations). Following the completion of 20 days of training, we tested mice for a 5-day block on each separation with 81 trials a day.

**Western blots and RT-PCR**. We isolated the dorsal dentate gyrus (dDG) from the Nestin$^{M1siRNA+}$ or AD/Nestin$^{M1siRNA}$ mice and their respective controls. The tissues were sliced and digested in buffer consisting of 10 mM Tris-Cl (pH 7.6), 50 mM NaF, 1 mM Na$_3$VO$_4$, 1 mM edetic acid, 1 mM benzamidine, 1 mM PMSF, 1 mg/10 ml papain, and a mixture of aprotinin, leupeptin, and pepstatin A (10 μg/ml each). After the 30-min digestion, we purified the suspended dNGIs$^{GFP+}$ cells from the digested tissues using an S3e Cell Sorter (Bio-Rad). The purified dNGIs$^{GFP+}$ cells were then homogenized and diluted with a buffer consisting of 200 mM Tris-Cl (pH 7.6), 8% SDS, 40% glycerol, 10% β-mercaptoethanol and 0.05% bromophenol blue at final concentration. We measured the protein concentration of the homogenized sample using a BCA kit (Pierce, Rockford, IL), as described before[26], [46, 57]. The samples were boiled for 10 min in a water bath. We separated the proteins in the extracts using 10% SDS-PAGE and transferred them to nitrocellulose membranes, which were then incubated with antibodies against M1 receptor (1:500, Epitomics, 3798–1), and M5 receptor (1:1000, abcam, ab41171), and beta-actin (1:2000, ab8227). The blots were scanned using an Infrared Imaging System (Odyssey, LI-COR) and quantitatively analyzed using Kodak Digital Science 1D software (Eastman Kodak, New Haven, CT).

For RT-PCR, we extracted total RNA from the purified dNGIs using TRIzol reagent according to the manufacturer's instructions (Sigma, St. Louis, MO), as previously described[46]. The primers for RT-PCR were as follows: M1 receptor: forward: 5′-TCCCTCACATCCTCCGAAGGTG-3′; reverse: 5′-CTTTCTTGGT GGGGCCTCTTGACTG-3; M5 receptor: forward: 5′-TAGCATGGCTGGTCTC CTTCA-3′, reverse: 5′-CGCTTCCCGACCAAGTACTG-3′. The PCR amplification was initiated with a 1-min denaturation step at 95 °C, denatured with 35 cycles at 95 °C for 10 s, annealed at 64 °C for 30 s, and extended at 72 °C for 60 s. The PCR products were separated by electrophoresis with 2% agarose consisting of 0.5 μg/ml ethidium bromide. The bands were visualized using a BioDoc-IT imaging system (Bio-Rad, Hercules, CA), and measured using a Bio-Rad GS-710 calibrated imaging densitometer.

**Statistical analysis**. All values in the text and Figure legends are represented as the mean ± SEM. All statistical results were shown in Supplementary Table 2. Parametric tests, including unpaired two-tailed Student's $t$ tests in electrophysiological recordings and post hoc Bonferreni's following two-way analyses of variance (ANOVA) in all behavioral tests were used when assumptions of normality and equal variance ($F$ test) were met. Significance was accepted for $p < 0.05$. Power calculations were performed using G*power software v3.1.9.2 (IDRE Research Technology Group, Los Angeles, USA). Group sizes were estimated based on recent studies[12, 26, 46] and were designed to provide at least 80% power with the following parameters: probability of type I error ($\alpha$) = 0.05, a conservative effect size of 0.25, and 3–8 treatment groups with multiple measurements obtained per replicate.

**Data availability**. All the relevant data supporting the findings of this study are available within the article and its Supplementary Information files or from the corresponding authors upon request.

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

## Acknowledgements

This work was supported by the National Natural Science Foundation of China (Grants: 91632306 to Y.L.; 51627807 to Y.L.; 31721002 to Y.L.; 31571039, 81771150 and 91632114 to L.-Q.Z.) and National Program for Support of Top-Notch Young Professionals, and Academic Frontier Youth Team of HUST to L.-Q.Z. We sincerely thank Ms. Na Wei (HUST) for behavioral analyses, Dr. Hui-Juan Jin (HUST) for mouse breeding and genotyping, Shan Wang (HUST) for Western blot analysis.

## Author contributions

Y.L., L.-Q.Z., and Y.M. conceived and designed the studies and wrote the paper. X.Y., T.T., and C.Y. performed optogenetics and electrophysiological studies. X.L., H.Z., L.P. and H.L. carried out synaptic tracing, immunohistochemistry, and cell counting. H.Z., W.C., Y.C. and S.S. performed both in vivo and in vitro electrophysiological recordings. H.Z., J.P., D.L., and M.-H.L. constructed virus vectors and generated mutant strains of mice. H.Y., X.L., and H.M. performed injections and imaging. All authors contributed to the data analysis and presentation in the paper.

## Additional information

**Competing interests:** The authors declare no competing financial interests.

