## [Peer Review File · Nature Communications]

Reviewers' comments:

Reviewer #1 (Remarks to the Author):

This is an impressive series of studies designed to explore the relationship between cholinergic innervation of new neurons in the dorsal hippocampus in spatial pattern separation and then to enhance this relationship in an effort to restore pattern separation function in an Alzheimer's disease mouse model. Using transgenic and viral vector approaches, the authors show that cholinergic neurons of the vertical limb of the diagonal band innervate new neurons in the dorsal dentate gyrus. They further show that activating this pathway increases the survival of new neurons in this area. Along with more new neurons, they show improved performance on pattern separation tasks. In an AD mouse model, they show impaired performance on pattern separation tasks and a reversal of this deficit by repetitive stimulation of cholinergic inputs as well as by activating new neurons through cholinergic receptors. Taken together, the results are compelling but there remain a few weaknesses in the paper. They are listed below in no order of importance:

- 1) The conclusion of the paper seems like an overstatement. While tapping into this mechanism may provide some clues about developing therapies for treatment of AD, it seems unlikely that the specific approach used here, which would be highly invasive, is a reasonable treatment choice for human patients, at least at this time. These claims should be tempered.
- 2) Along these lines, given the reduction in adult neurogenesis that occurs with aging, it is questionable whether approaches that target new neurons in elderly human populations would be suitable. Age-related decline in adult neurogenesis should be discussed in the paper.
- 3) The distinction between activation of new neurons and enhancement of cell survival by cholinergic innervation should be explored further. While it is likely that these two phenomena are related, exploring the ability of cholinergic stimulation to enhance pattern separation in the absence of an increase of new neurons (by using transgenic mice lacking neurogenesis or focal irradiation) would add to the paper.
- 4) Only one AD mouse model was used. The results would be strengthened by replicating at least some of the findings in another model.
- 5) The connection between the specificity of cholinergic stimulation and increased survival of new neurons should be explored further. Although the authors have shown that new neurons receive cholinergic innervation from these studies, it remains unknown whether increasing input from other afferents would have a similar effect. These results would be further strengthened by exploring the specificity of the cholinergic afferents in inducing increased cell survival.
- 6) The photographic images of BrdU labeling in Figure 3b are difficult to see and considerably lower quality than the other images in the paper.
- 7) The graphs would be enhanced by including the actual data points on the bars so that the variability and n size for each group is evident.

Reviewer #2 (Remarks to the Author):

In this study, Huanhuan Yan et al have extensively investigated roles of vChATs-dNGIs synaptic transmission in adult hippocampal neurogenesis and spatial pattern separation in mice. The authors found that that vChATs directly innervate NGIs in the dorsal zone of the hippocampus in adult mice using a genetically modified Cre-dependent anterograde monosynaptic tracing system. They also showed that vChATs-dNGIs synaptic transmission enhances adult hippocampal neurogenesis by examining effects of optogenetic activation and inactivation of vChATs-dNGIs synaptic transmission. Furthermore, they have shown that increased or reduced hippocampal neurogenesis is reflected by improved and impaired, respectively, spatial pattern separation at the behavioral levels. Finally, they have shown that AD model mice show impairments in vChATs-dNGIs synaptic transmission, that this impairment is associated with the defects of spatial pattern separation and that activation of vChATs using a theta burst stimulation paradigm rescues the

impairments of spatial pattern separation and neurogenesis. The experiments were well designed and the results look so clear and interesting. Additionally, the authors used elegant and excellent techniques. However, I have several major concerns as follows.

- 1) The authors should extensively revise the manuscript. There are much redundancy (such as copy and paste) and mistakes. I also recommend that the authors should ask English editor to correct English of some parts in this manuscript.
- 2) Importantly, what are main findings of this manuscript? Title and abstract seem to reflect the results in Figure 5. However, the manuscript contains additional interesting findings. The authors should revise the title and abstract.
- 3) Figure 5; I agree that AD mice show impaired synaptic transmission of vChATs-dNGIs. However, TBS improved NPA, spatial pattern separation and neurogenesis not only in AD mice also in WT mice, suggesting that effects of TBS are not specific for AD mice. From this view, the title and abstract seem to contain overstatements.
- 4) TBS failed to improve deficits in spatial (water maze) and object recognition memories in AD mice although this improved an impairment of spatial pattern separation observed in AD mice (this is a interesting finding). From this view, it is not sure whether activation of vChAT is a great strategy to improve AD.
- 5) Statistic analyses; The results of two way ANOVA are missing in many parts in the Results session. The authors should show all of the results. It may be good to summarize in the Table. More importantly, why did the authors perform student's t-tests following ANOVA? I recommend to perform, for example, post hoc Bonferroni's comparisons following two way ANOVA

Minor comments:

- 1) No reference characterizing ChATs-CreGFP mice and Nestin-Cre ER mice (they showed only stock numbers in the Jackson).
- 2) The title of "Single-Cell Western and RT-PCR" in the Methods is problematic. They did not perform western blotting and RT-PCR at the single cell level.

Reviewer #3 (Remarks to the Author):

Yan, and collaborators studied which neurons in the hippocampus are innervated by cholinergic projection neurons in mice. They suggest there is a cholinergic projection from the diagonal band of Broca to the Dentate Gyrus (DG), more particularly, that cholinergic neurons innervate selectively newly generated immature neurons in the Dentate Gyrus. Using sophisticated mouse lines and paradigms they suggest there are important implications of cholinergic tone for survival of new neurons and their involvement in spatial pattern separation. they suggest that this observations are also relevant in a mouse model of AD.

Using retrograde viral labeling and expression of ChR2, the authors show they can specifically activate or inactivate these cholinergic projection neurons that innervate the immature neuronal population. They go on to record from these newly generated neurons and found a M1 muscarinic sensitive related current elicited by activation of cholinergic neurons with optogenetics. These currents peak 15 days after viral injection. They then use a stimulation protocol (theta burst, TBS) for 16 days once a day and examined neurogenesis by injecting the mice with BrdU and using a pulse chase approach. They find that survival of BrdU labeled neurons have a different kinetics between TBS and non-TBS mice, and in mice that received TBS, BrdU neurons after they peak in numbers, had a smaller decrease when compared to controls, likely because survival was improved. Complementary experiments using optogenetics to silence cholinergic tone 15 min/day indicate that this later protocol decrease the survival of BrdU labeled new neurons.

Surprisingly, these two protocols seem to regulate behavior, specifically in a task to measure pattern separation using touchscreens, performed almost 5 months after either manipulation

protocol. The authors go on to generate AD mice in which they can manipulate the expression of ChR2 in new neurons and their cholinergic connections. For that, they generated a complex animal model Tg2576 (homos) expressing Cre and TVA receptor and rabies G ; all of them homozygous.

They show that the AD mice have decreased number of cholinergic terminals contacting these new immature neurons in the DG; they then demonstrate AD mice have decreased levels of BrdU positive cells and both MWM and pattern separation deficits. The activation of cholinergic neurons with theta bursts for a bit more than 26 days (once a day) also caused a recovery of BrdU positive cells and rescued the deficits in pattern separation tested about 5 months after the stimulation.

The questions asked are of importance and the amount of work in this manuscript is remarkable. The conclusions that cholinergic neurons innervate newly born neurons is of interest to the field. The manuscript has a number sophisticated experiments. However, I have several issues with the data and some of the key experiments that decrease my enthusiasm for the manuscript significantly.

Overall the manuscript needs improvement in several important aspects. Material and methods is very confusing with numerous acronyms and a large number of transgenes that becomes very confusing. Please refrain to use so many abbreviations and consider a Figure illustrating the mouse lines used. Some of the figures do not have information in the material and methods session e.g., Fig 2e. there is no mention in the text about the compounds used even though they state they tested different muscarinic receptors. Hence it becomes hard to follow the experiments performed. There are misspelling and repeated sentences (particularly, in Fig legends) as well as some odd sentences (for ex rescues the impairments of spatial pattern separation by intercepting the decays of cholinergic synaptic transmission).

I have also a number of issues with the data:

Fig 1: Please provide quantifications for double-labeled neurons with proper statistics. For the main conclusion in Fig 1H is imperative to know that there is no other neuronal type in addition to cholinergic neurons that gets labeled as well. So please provide quantification for these data.

Fig 2a. The representation of the firing pattern does not make sense like this. Please, show a baseline and the moment of the stimulation with blue light and the stimulation protocol. Also quantification of double-labeled neurons should be done with proper statistical variation.

Fig 2. Why do the currents peak and then decrease? Would this affect the in vivo experiments? What is the basis of the muscarinic-activated current in these neurons? Atropine or M1 are not named in Materials and Methods. Also, is not clear what Ni stands for or the different Muscarinic compounds used in this experiment. Please, explain these.

Fig 3a. Please provide the exact protocol for the theta burst stimulation. Is this 4x 100 Hz? When was the theta stimulation performed after the injection of virus? You show that muscarinic responses peak at day 16 after virus injections, how this relates to muscarinic responses in vivo?. Can you quantify the responses of target neurons for the theta stimulation? Was the recording done every time the brain was stimulated?

The examples of BrdU labeling are very hard to see; please provide better examples with improved contrast that allows one to confirm the quantified labeling.

Fig 3a-f Conclusion: TBS stimulation increase survival, or delays maturation of NGIs, it does not induce neurogenesis per se, that has not been shown here; the number of BrdU+ cells does not increase at day 7. Same with the inactivation of vChATs, it inhibits dNGI survival, it does not impair neurogenesis. In a number of sentences the authors state that these protocols increase or decrease neurogenesis. Also cholinergic transmission is not essential; it regulates the process.

Fig 3g. Student's t-test is not the appropriate statistical analysis in this experiment. You should use ANOVA. And you should add all the statistical information on the manuscript (F values). Moreover, for several of the behavior data you state that ANOVA was used for the analysis, but in the figure legend you mention t-test. Which one was actually done?

For the inhibition and the stimulation protocols, how did you choose them? It is quite surprising that only 15 min silencing of cholinergic neurons once a day has an effect in the survival of newly generated neurons in the DG.

What is the relationship between NeuN/GFAP positive cells and BrdU labeled cells?
Why in supplementary fig 2 you show only GFAP+ cells instead of GFAP/NeuN cells?

Supplementary Fig 4b. You are not showing the % of time spent on novel versus familiar objects. Is the Time % shown here the total exploration time? Or it should be RI?

You did not do single cell Western or PCR. A population of cells was isolated. Please run gels with at least 3 samples for each condition to demonstrate reproducibility. One sample is just not adequate.

On line 212, the statement that cholinergic transmission is necessary and sufficient for neurogenesis is an overstatement.

The protocols for behavior in Fig 4 are not clear. You state that behavior was done 1 week after the TBS stimulation was finished. Are all animals going through all behavior tests? Or are they separate groups? When did you inject BrdU? Without a clear explanation on how the behavioral experiments have been performed, a conclusion cannot be made regarding their validation.

It seems that animals were run in touchscreens after the stimulation. Training and performance of PAL takes almost 3 months, and most likely 1.5 months for pattern separation using location discrimination. It is difficult to come to terms that the short period of stimulation during 16 days has such a prolonged effect almost 5 months after finishing the stimulation.

For touchscreen experiments you have acquired other sets of data. Correction errors and time to collect reward and other parameters should be shown for PAL and LD (you did not do TUNEL please correct material and methods) as supplementary data. I am also surprised with the protocol you used to motivate mice to perform touchscreen tasks. C57 mice are usually kept at 85% of their original weight, which is used to motivate mice to do touchscreen tests. Is it correct that you can start the experiments by reducing food to 3g per mouse per day. Are the mice isolated? We and others usually reduce slowly the weight over a week and maintain their weight at 85% of original weight using 1.5 to 2 g of food a day. I suspect that 3 g would be more than enough to keep the mice fed. Please explain this difference in protocol when compared to others. t Test is not appropriate for analysis of PAL results.

For all the BrdU correlations provide stats and R². When was BrdU injected for these experiments?

Figs 4 f and i. A lot of the data seem identical in these two graphs. Are they the same data? Please also provide analysis of other key parameters in supplementary data.

Fig 5 and correspondent supplementary data: What is the rationale behind using a 28-days TBS while you were using a 16-days TBS in the rest of the paper? Please show representative images for the ChAT positive terminals in the two conditions matching the quantification.

Please use the same names for the mouse lines in the paper and try to make it easier to follow. For your AD/vChATsChR2+ mice there will be 3 different transgenes all homozygous. Please provide a short explanation of your breeding strategy to generate controls.

For the touchscreen tasks in the AD mice, what was the age the mice were actually trained and tested. Again, it takes 2 to 3 months to train for PAL, and 1.5 for LD, what is the exact age they were tested? I am surprised there are no deficits in PAL, even though these mice at 180 days have deficits in the MWM. Are all other parameters identical for AD mice? Please provide key experimental data for PAL and LD for these experiments as supplementary data (correction errors, time to collect rewards etc).

How does TBS treatment recover synaptic transmission? Please explain what the TBS treatment is (frequency and time of stimulation). I am again amazed that TBS for 15 days has lasting effects, both on the survival of nerve terminals and muscarinic currents. Please provide a potential mechanism for this. I am also amazed that the TBS stimulation for 28 days (why was the protocol changed?) has such long-term effects in the AD mice improving behavior deficits almost 5 months after finishing the stimulation. This is a very surprising result and very difficult to explain without understanding the theta stimulation parameters. Would survival of BrdU neurons not be compromised by the expression of mutated APP after the end of the stimulation? This needs to be discussed.

Please check mislabeling for Fig 5g.

T

the references for power analysis are not at all related to the behavior tests. How was then power calculated in these experiments?

The concept that acetylcholine, via M1 muscarinic receptors regulates neurogenesis, is not new (see *Neuropharmacology*. 2010 May;58(6):921-9. doi: 10.1016/j.neuropharm.2009.12.005. Epub 2009 Dec 22. Agonist-induced restoration of hippocampal neurogenesis and cognitive improvement in a model of cholinergic denervation. Van Kampen JM1, Eckman CB. this is just one example, there are several other publications). Please make sure to reference these initial works adequately.

Discussion: last paragraph seems a bit pretentious and doesn't really explain how this strategy could be implemented in AD patients. It mixes actual therapeutically approaches with techniques used in basic research. With all the information on this manuscript, I think the discussion could be richer. Please avoid odd sentences.

Replies to the reviewers

To Reviewer 1:

We greatly appreciate the reviewer's comments and suggestions. We feel that all comments raised by the reviewer are fair and constructive and these have been now fully addressed on a point-to-point basis in this revision.

- 1. Reviewer's comments:** The conclusion of the paper seems like an overstatement. While tapping into this mechanism may provide some clues about developing therapies for treatment of AD, it seems unlikely that the specific approach used here, which would be highly invasive, is a reasonable treatment choice for human patients, at least at this time. These claims should be tempered.

Our replies: As suggested by the reviewer, we have now changed the titles of this revision as the following "Impairments of spatial pattern separation in Alzheimer's disease via degeneration of cholinergic synapses in the hippocampus"

- 2. Reviewer's comments:** Along these lines, given the reduction in adult neurogenesis that occurs with aging, it is questionable whether approaches that target new neurons in elderly human populations would be suitable. Age-related decline in adult neurogenesis should be discussed in the paper.

Our replies: As suggested by the reviewer, we have now fully discussed neurogenesis in AD mice versus normal aging mice in the text of p20-L4-23.

- 3. Reviewer's comments:** The distinction between activation of new neurons and enhancement of cell survival by cholinergic innervation should be explored further. While it is likely that these two phenomena are related, exploring the ability of cholinergic stimulation to enhance pattern separation in the absence of an increase of new neurons (by using transgenic mice lacking neurogenesis or focal irradiation) would add to the paper.

Our replies: As suggested by the reviewer, we have now represented new data in new Fig 4g-j and new Supplementary Fig. 6C and described in the text of p12-L19-p13-L2 as the following: "This approach effectively inhibited both spatial pattern separation (Fig. 4e) and BrdU labeling (Fig. 4f). The similar inhibition was achieved by silencing M1 receptor gene in dNGIs (Fig. 4g-j). Significantly, we found that silencing the M1 receptor gene that blocks cholinergic synaptic transmission in dNGIs and reduces the dNGIs survival completely eliminated the enhancement of spatial pattern separation induced by TBS (Fig. 4j and Supplementary Fig. 6C). Thus, cholinergic transmission in dNGIs controls spatial pattern separation via the regulation of the dNGIs survival."

- 4. Reviewer's comments:** Only one AD mouse model was used. The results would be strengthened by replicating at least some of the findings in another model.

Our replies: As suggested by the reviewer, we have now used an additional mouse model of AD, in which both the APPKM670/671NL(Swedish) and the PSEN1deltaE9 mutant genes were expressed (AD1 mice). We crossed AD1 mice with the vChATs^{Chr2+}, resulting in the AD1/vChATs^{Chr2+} mice. These new data have been now represented in new Supplementary Fig. 13, and described in the text of p17-L12-24, as the following: "The roles of cholinergic synaptic degeneration in the spatial memory loss were further investigated by using an additional mouse model of AD, in which both the APPKM670/671NL(Swedish) and the PSEN1deltaE9 mutant genes were expressed (AD1

mice). We crossed AD1 mice with the $vChATs^{ChR2+}$, resulting in the $AD1/vChATs^{ChR2+}$ mice (Supplementary Fig 12A). In a low separation of 2-CSD task, the performance of $AD1/vChATs^{ChR2+}$ mice was significantly reduced, compared to the age-matched control/ $vChATs^{ChR2+}$ mice (Supplementary Fig. 13 B,C) and this reduction of the performance was associated with a decrease of BrdU labeling in the dorsal portion of the hippocampus (Supplementary Fig. 13D). Significantly, TBS treatment effectively improved the pattern-separation-associated spatial memory (Supplementary Fig. 13E). These data using an additional AD mouse model confirm that the capacity of the dNGIs survival is reduced and this reduction of the dNGIs survival contributes to the impairments of spatial pattern separation in AD mice”.

- 5. Reviewer’s comments:** The connection between the specificity of cholinergic stimulation and increased survival of new neurons should be explored further. Although the authors have shown that new neurons receive cholinergic innervation from these studies, it remains unknown whether increasing input from other afferents would have a similar effect. These results would be further strengthened by exploring the specificity of the cholinergic afferents in inducing increased cell survival.

Our replies: As suggested by the reviewer, we have now determined the specificity of cholinergic activation in the dNGIs survival. We stimulated excitatory synaptic inputs from pyramidal neurons in the entorhinal cortex (EC_{PN}). We used the EC_{PN}^{ChR2+} mice, in which ChR2 was expressed in excitatory pyramidal neurons. TBS was delivered through an optical fiber located in the EC layer II region of the EC_{PN}^{ChR2+} mice once per day for 16 consecutive days. BrdU⁺ cells in the dorsal hippocampus were then examined. Our data showed that TBS activation of EC_{PN} did not alter the survival of dNGIs in adult mice (new Supplementary Fig.4A-D) and described in the text of p8-117-23 in this revision.

- 6. Reviewer’s comments:** The photographic images of BrdU labeling in Figure 3b are difficult to see and considerably lower quality than the other images in the paper.

Our replies: As suggested by the reviewer, we have now represented high quality images of BrdU labeling in new Fig 3b in this revision.

- 7.** The graphs would be enhanced by including the actual data points on the bars so that the variability and n size for each group is evident.

Our replies: As suggested by the reviewer, we have now represented the actual data in all bar graphs throughout the manuscript including new Figs 1-5 and new Supplementary Figs 1-13.

To Reviewer 2:

We greatly appreciate the reviewer's comments and suggestions. We feel that all comments raised by the reviewer are fair and constructive and these have been now fully addressed on a point-to-point basis in this revision.

1. **Reviewer's comments:** The authors should extensively revise the manuscript. There are much redundancy (such as copy and paste) and mistakes. I also recommend that the authors should ask English editor to correct English of some parts in this manuscript.

Our replies: As suggested by the reviewer, we have now extensively revised the manuscript. The original manuscript has been now edited by 3 English editors.

2. **Reviewer's comments:** Importantly, what are main findings of this manuscript? Title and abstract seem to reflect the results in Figure 5. However, the manuscript contains additional interesting findings. The authors should revise the title and abstract.

Our replies: As suggested by the reviewer, we have now changed the title of the manuscript as the following: "**Impairments of spatial pattern separation in Alzheimer's disease via degeneration of cholinergic synapses in the hippocampus**". The abstract has been now revised as the following; "...Activation of vChATs with the theta burst stimulation (TBS) that intercepts the decays of cholinergic synaptic transmission effectively protects against the declines of spatial pattern separation and this protection was completely abolished by inhibiting the dNGIs survival. Thus, the impairments of pattern separation-associated spatial memory are caused by degeneration of cholinergic synaptic transmission that mediates the dNGIs survival in AD mice. "

3. **Reviewer's comments:** Figure 5; I agree that AD mice show impaired synaptic transmission of vChATs-dNGIs. However, TBS improved NPA, spatial pattern separation and neurogenesis not only in AD mice also in WT mice, suggesting that effects of TBS are not specific for AD mice. From this view, the title and abstract seem to contain overstatements.

Our replies: As suggested by the reviewer, we have now changed the title of the manuscript as the following: "Impairments of spatial pattern separation in Alzheimer's disease via degeneration of cholinergic synapses in the hippocampus". The abstract has been now revised as the following; "Activation of vChATs with the theta burst stimulation (TBS) that intercepts the decays of cholinergic synaptic transmission effectively protects against the declines of spatial pattern separation and this protection was completely abolished by inhibiting the dNGIs survival. Thus, the impairments of pattern separation-associated spatial memory are caused by degeneration of cholinergic synaptic transmission that mediates the dNGIs survival in AD mice".

4. **Reviewer's comments:** TBS failed to improve deficits in spatial (water maze) and object recognition memories in AD mice although this improved an impairment of spatial pattern separation observed in AD mice (this is an interesting finding). From this view, it is not sure whether activation of vChAT is a great strategy to improve AD.

Our replies: As suggested by the reviewer, we have now removed a sentence "...this is a promising strategy for AD therapy..." throughout the text in this revision.

5. **Reviewer's comments:** Statistic analyses; the results of two-way ANOVA are missing in many parts in the Results session. The authors should show all of the results. It may be good to summarize in the Table. More importantly, why did the authors perform student's

t-tests following ANOVA? I recommend to perform, for example, post hoc Bonferroni's comparisons following two way ANOVA

Our replies: As suggested by the reviewer, we have now performed statistics using post hoc Bonferroni's comparisons following two-way ANOVA, as shown in the Figure legends and described in the section of statistical analysis of this revision.

Minor comments:

1. **Reviewer's comments:** No reference characterizing ChATs-CreGFP mice and Nestin-Cre ER mice (they showed only stock numbers in the Jackson).

Replies: As suggested by the reviewer, we have now described the mutant mice with both stock and reference numbers in the section of Methods of this revision.

2. **Reviewer's comments:** The title of "Single-Cell Western and RT-PCR" in the Methods is problematic. They did not perform western blotting and RT-PCR at the single cell level.

Replies: As suggested by the reviewer, we have now changed the title of "Single-Cell Western and RT-PCR" into "Western and RT-PCR" in the Methods in this revision

To Reviewer 3:

We greatly appreciate the reviewer's comments and suggestions. We feel that all comments raised by the reviewer are fair and constructive and these have been now fully addressed on a point-to-point basis in this revision.

1. **Reviewer's comments:** Overall the manuscript needs improvement in several important aspects. Material and methods is very confusing with numerous acronyms and a large number of transgenes that becomes very confusing. Please refrain to use so many abbreviations and consider a Figure illustrating the mouse lines used. Some of the figures do not have information in the material and methods session e.g., Fig 2e. there is no mention in the text about the compounds used even though they state they tested different muscarinic receptors. Hence it becomes hard to follow the experiments performed. There are misspelling and repeated sentences (particularly, in Fig legends) as well as some odd sentences (for ex rescues the impairments of spatial pattern separation by intercepting the decays of cholinergic synaptic transmission).

Our replies: As suggested by the reviewer, we have now extensively revised the manuscript by 3 English editors. For the clarity, we have now listed the mutant lines of mice in new Supplementary Table 1. We have described the compounds in the section of Methods (p24-L1-17), and also in new Fig.2e legend.

2. **Reviewer's comments:** Fig 1: Please provide quantifications for double-labeled neurons with proper statistics. For the main conclusion in Fig 1H is imperative to know that there is no other neuronal type in addition to cholinergic neurons that gets labeled as well. So please provide quantification for these data.

Our replies: As suggested by the reviewer, we have now represented actual and averaged numbers of doubled labeled cells in the dorsal dentate gyrus in new Supplementary Fig. 1A and vDB in new Supplementary Fig. 1B and entorhinal cortex in new Supplementary Fig. 1C.

3. **Reviewer's comments:** Fig 2a. The representation of the firing pattern does not make sense like this. Please, show a baseline and the moment of the stimulation with blue light and the stimulation protocol. Also quantification of double-labeled neurons should be done with proper statistical variation.

Our replies: As suggested by the reviewer, we have now shown new representative *in vivo* recordings (instead of the original *in vitro* recordings) of the firing pattern evoked by blue laser lights in new Fig.2a, and the actual and average numbers of double labeled ChR2 and ChATs have been now represented in new Supplementary Fig. 2.

4. **Reviewer's comments:** Fig 2. Why do the currents peak and then decrease? Would this affect the *in vivo* experiments? What is the basis of the muscarinic-activated current in these neurons? Atropine or M1 are not named in Materials and Methods. Also, is not clear what Ni stands for or the different Muscarinic compounds used in this experiment. Please, explain these.

Our replies: As suggested by the reviewer, we have now described atropine and M1 receptor inhibitor in new Fig. 2 legend as well as in the section of Methods in p24-L1-7, as the following: "The currents were sensitive to the general muscarinic acetylcholine receptor antagonist atropine (5 μ M) and a selective type-1 muscarinic (M1) acetylcholine receptor antagonist VU0255035 (5 μ M, Fig. 2e), but not to the nicotinic acetylcholine

receptor antagonist (Ni) and the type-2 (M2), type-3 (M3 and type-4 (M4) of the muscarinic acetylcholine receptor antagonists". The declines of peak currents that mediated by activation of type 1 of muscarinic acetylcholine receptor have been now discussed in the text of p6-L23-p7-11 in this revision as the following: "...suggesting that vChATs establish the functional synaptic connections with dNGIs only, but not with the matured granule cells in the dorsal hippocampus".

5. **Reviewer's comments:** Fig 3a. Please provide the exact protocol for the theta burst stimulation. Is this 4x 100 Hz? When was the theta stimulation performed after the injection of virus? You show that muscarinic responses peak at day 16 after virus injections, how this relates to muscarinic responses *in vivo*? Can you quantify the responses of target neurons for the theta stimulation? Was the recording done every time the brain was stimulated? The examples of BrdU labeling are very hard to see; please provide better examples with improved contrast that allows one to confirm the quantified labeling.

Our replies: As suggested by the reviewer, we have now described the TBS protocol used in this study in the section of Methods in this revision of p25-L1-4 as the following: "TBS consists of 10 trains of stimuli at 10 seconds intervals, with each train containing bursts of 4 spikes at 100 Hz and repeated 10 times at 5 Hz and was applied for 16 consecutive days (16-days TBS) in normal adult mice and for 28 consecutive days (28-days TBS) in AD mice.". The experimental schedule for TBS and BrdU injection and BrdU labeling were described in new Fig. 3a and new Supplementary Fig. 3a legends, respectively, as the following: "Experimental schedule shows that mice at 120 ± 2 days old of age were administered with a single dose of BrdU. 7 days after the BrdU administration (DAI), TBS was applied for 16 consecutive days. BrdU staining was then performed 1, 3, 7, 11, 21, or 28 at DAI", "Experimental schedule shows that adult mice at 120 ± 2 days old of age were applied with 16-day TBS. One day before end of TBS application. A single dose of BrdU was administered. BrdU labeling was done 1, 6, 12, 18 or 24 hrs after the BrdU injection (HAI)".

Our data in Fig. 2c showed that vChATs establish the functional synaptic connections with dNGIs beginning 7 days after the birth. Based on this finding, TBS protocol was applied for 16 consecutive days into the mice at 7 DAI, as described in Fig. 3a and new Supplementary Fig. 3A legends. Yes, we performed multiple electrode recordings in the dorsal dentate gyrus of mice *in vivo* when TBS was applied. But, we cannot quantify the responses of the individual dNGIs because the intracellular recordings from the individual dNGIs of mice *in vivo* are not applicable.

6. **Reviewer's comments:** Fig 3a-f Conclusion: TBS stimulation increase survival, or delays maturation of NGIs, it does not induce neurogenesis per se, that has not been shown here; the number of BrdU⁺ cells does not increase at day 7. Same with the inactivation of vChATs, it inhibits dNGI survival, it does not impair neurogenesis. In a number of sentences the authors state that these protocols increase or decrease neurogenesis. Also cholinergic transmission is not essential; it regulates the process.

Our replies: As suggested by the reviewer, we have now changed the neurogenesis in the original manuscript into the dNGIs survival throughout the text in this revision.

7. **Reviewer's comments:** Fig 3g. Students t-test is not the appropriate statistic analysis in this experiment. You should use ANOVA. And you should add all the stat information on the manuscript (F values). Moreover, for several of the behavior data you state that ANOVA was used for the analysis, but in the figure legend you mention t-test. Which one was actually done?
For the inhibition and the stimulation protocols, how did you choose them? It is quite

surprising that only 15 min silencing of cholinergic neurons once a day has an effect in the survival of newly generated neurons in the DG.

Our replies: As suggested by the reviewer, we have now performed two-way ANOVA analysis in all behavioral studies and showed both *F* and *P* values in the figure legends and the results section in the text of this revision. In our earlier studies, we have tested three different protocols for the inhibition of cholinergic synaptic transmission and the dNGIs survival. One protocol consists of two stimuli (each lasts for 15 min) a day at 12 hours interval for 6 consecutive days; the second protocol consists of one stimulus (lasts for 15 min) a day for 12 consecutive days, and the third protocol consists of one stimulus (lasts for 15 min) a day for 16 consecutive days. We have found that 15 min silencing of cholinergic synaptic transmission once a day for 16 consecutive days is effective for inhibiting the dNGIs survival, and hence we used this protocol for the essential roles of cholinergic synaptic transmission in the dNGIs survival.

8. **Reviewer's comments:** What is the relationship between NeuN/GFAP positive cells and BrdU labeled cells? Why in supplementary fig 2 you show only GFAP+ cells instead GFAP/NeuN cells

Our replies: As suggested by the reviewer, we have now represented the new images of the sections that were labeled with BrdU, GFAP and NeuN in the new Supplementary Fig. 5A. We have also represented new data in the new Supplementary Fig. 5B showing that numbers of BrdU⁺GFAP⁺ and BrdU⁺NeuN⁺ cells.

9. **Reviewer's comments:** Supplementary Fig 4b. You are not showing the % of time spent on novel versus familiar objects. Is the Time % showed here the total exploration time? Or it should be RI.

Our replies: As suggested by the reviewer, we have now represented new data in new Supplementary Fig. 7 showing the Time% spent on novel versus familiar objects.

10. **Reviewer's comments:** You did not do single cell Western or PCR. A population of cells was isolated. Please run gels with at least 3 samples for each condition to demonstrate reproducibility. One sample is just not adequate.

Our replies: As suggested by the reviewer, we have now changed the Methods of "Single cell Western and PCR" into "Western blots and RT-PCR" in this revision. We have also represented new images of gels with 3 samples for each condition in new Fig.3g.

11. **Reviewer's comments:** On line 212, the statement that cholinergic transmission is necessary and sufficient for neurogenesis is an overstatement

Our replies: As suggested by the reviewer, we have now changed this sentence as the following: "cholinergic transmission between vChATs and dNGIs plays an essential role in the survival of dNGIs in adult mice" in this revision of p11-11-2.

12. **Reviewer's comments:** The protocols for behavior in Fig 4 are not clear. You state that behavior was done 1 week after the TBS stimulation was finished. Are all animals going through all behavior tests? Or are they separate groups? When did you inject BrdU? Without a clear explanation on how the behavioral experiments have been performed, a conclusion cannot be made regarding their validation.

Our replies: As suggested by the reviewer, we have now represented the experimental schedule on each panel of the graphs in new Fig. 4. In PAL tests (Fig. 4a), the mice at

120 ± 2 days old of age were treated with 16 consecutive days of TBS. 7 days after the beginning of the TBS treatment, mice were performed with a PAL test for 60 days. In the PAL tests, mice were not administered with BrdU and were not performed with BrdU staining. “In a 2-CSD task (Fig. 4b), mice (after the completion of PAL) were performed with PAL tests for 60 days. After the completion of PAL tests, mice were administered with a single dose of BrdU. 7 days after the BrdU administration, mice were trained with 2-CSD. 4 days after the beginning of the training, mice were treated with 16 consecutive days of TBS. Following the completion of 20 days of the training, the mice were applied for a 5-day block test on each separation (low or high). Mice were required to complete 81 trials a day. Immediately after end of the tests, mice were killed for BrdU staining”. This has been now described in Fig. 4b legend.

13. **Reviewer’s comments:** It seems that animals were run in touchscreens after the stimulation. Training and performance of PAL takes almost 3 months, and most likely 1.5 months for pattern separation using location discrimination. It is difficult to come to terms that the short period of stimulation during 16 days has such a prolonged effects almost 5 months after finishing the stimulation.

Our replies: As suggested by the reviewer, we have now represented the experimental arrangements that show the schedules in PAL and 2-CSD tests in new Fig.4a, Fig. 4b. These schedules were described in new Fig.4a and Fig. 4b legends in this revision.

14. **Reviewer’s comments:** For touchscreen experiments you have acquired other sets of data. Correction errors and time to collect reward and other parameters should be shown for PAL and LD (you did not do TUNL please correct material and methods) as supplementary data. I am also surprised with the protocol you used to motivate mice to perform touchscreen tasks. C57 mice are usually kept at 85% of their original weight, which is used to motivate mice to do touchscreen tests. Is it correct that you can start the experiments by reducing food to 3/g per mice per day. Are the mice isolated? We and others usually reduce slowly the weight over a week and maintain their weight at 85% of original weight using 1.5 to 2 g of food a day. I suspect that 3 g would be more than enough to keep the mice feed. Please explain this difference in protocol when compared to others. t Test is not appropriate for analysis of PAL results.

Our replies: I completely agree with the reviewer’s comments that we did not clearly describe the experimental procedures in the performance of touchscreen tasks. Methods have been now corrected based on our experimental procedures in this revision of p27L19-p29L5. In the present studies, sugar pills were used as a reward. Food was restricted to maintain 85-90% of free feeding body weight. As suggested by the reviewer, we have now represented new data in new Supplementary Fig.10 showing the numbers of bean breaks, screen touches and errors% during the pre-training sessions. Two-way ANOVA was used for the analysis of the behavioral results, as described in the section of Methods and new Figure legends.

15. **Reviewer’s comments:** For all the BrdU correlations provide stats and R2. When was BrdU injected for these experiments?

Our replies: As suggested by the reviewer, we have now described the experimental procedures for the BrdU administration and labeling in all new figure legends in this revision and Rsqr values have been now shown in new Figs 4c-g legends.

16. **Reviewer’s comments:** Figs 4 f and i. A lot of the data seem identical in these two graphs. Are they the same data? Please also provide analysis of other key parameters in supplementary data.

Our replies: We greatly appreciate the reviewer's comments on the original Fig. 4 f and i. For the clarity, we have now represented the actual and averaged new data in new Fig. 4h and i, and described the key parameters in the figure legends of this revision, as the following: "Silencing M1 receptor gene blocks the beneficial effects of TBS on spatial pattern separation. Plots show the actual (blue circles) and mean \pm SEM (red triangles) of the trials that are required to reach a criterion at a low (**h**) or high (**i**) separation from the individual animals ($n = 11$ mice/group, $F_{1,43} = 9.79$, $*p = 0.0012$, $**p = 0.00082$, two-way ANOVA). (**j**) The numbers of trials that are required to reach a criterion at a low separation in Fig 4h are plotted against the number of BrdU⁺ cells from the individual animals that were treated with TBS".

17. **Reviewer's comments:** Fig 5 and correspondent supplementary data: What is the rationale behind of using a 28-days TBS while you were using a 16-days TBS in the rest of the paper? Please show representative images for the ChAT positive terminals in the two conditions matching the quantification.

Our replies: In our previous studies, we have tested several protocols similar to these used in deep brain stimulation in the human patients (references 34; 42-44). In our recent report (Yang et al., Molecular Psychiatry 2016, ref 18), we have shown that 28-day TBS effectively prevents synaptic degeneration AD mice. Accordingly, we have applied the same protocol that was used before in the present studies, aiming to protect against cholinergic synaptic decays in AD mice. As suggested by the reviewer, we have now represented an additional image showing the ChAT-positive terminals in AD mice treated with TBS in new Fig. 5b.

18. **Reviewer's comments:** Please use the same names for the mouse lines in the paper and try to make it easier to follow. For your AD/vChATs^{Chr2+} mice there will be 3 different transgenes all homozygous. Please provide a short explanation of your breeding strategy to generate controls.

Our replies: As suggested by the reviewer, we have now listed all transgenic lines in new Supplementary Fig. 14, and described the strategies for generation of the individual control mice in the new Supplementary Fig. 14 as the following: "AD mice were crossed with vChATs^{Chr2+} mice, resulting in both AD/vChATs^{Chr2+} and the non-transgenic control/vChATs^{Chr2+} mice and control/vChATs^{Chr2+} from the same litter of AD/vChATs^{Chr2+} mice were used as the controls".

19. **Reviewer's comments:** For the touchscreen tasks in the AD mice, what was the age the mice were actually trained and tested. Again, it takes 2 to 3 months to train for PAL, and 1.5 for LD, what is the exact age they were tested? I am surprised there are no deficits in PAL, even though these mice at 180 days have deficits in the MWM. Are all other parameters identical for AD mice? Please provide key experimental data for PAL and LD for these experiments as supplementary data (correction errors, time to collect rewards etc).

Our replies: As suggested by the reviewer, we have now shown the experimental schedules in new Fig. 5a, e, f and g, and described the experimental schedules in the respective legends in this revision. For clarity, we have now represented the data of PAL in new Supplementary Fig. 12. Our data in both the present studies and the previous reports (Yang et al., Molecular Psychiatry 2016; Shu et al., Journal of Neuroscience 2016) have shown that AD mice had the deficits of spatial memory in both 2-CSD and MWM tests and these mice were normal in all other behavioral tests including open field tests, as shown in new Supplementary Fig. 11. Our results are consistent with the most previous reports that the spatial memory loss is an earliest behavioral sign in AD mice

and overall behavioral deficits including motor and social activities are found only when they are at 8 months old of age or older.

20. **Reviewer's comments:** How does TBS treatment recover synaptic transmission? Please explain what the TBS treatment is (frequency and time of stimulation). I am again amazed that TBS for 15 days has lasting effects, both on the survival of nerve terminals and muscarinic currents. Please provide a potential mechanism for this. I am also amazed that the TBS stimulation for 28 days (why was the protocol changed?) has such long-term effects in the AD mice improving behavior deficits almost 5 months after finishing the stimulation. This is a very surprising result and very difficult to explain without understanding the theta stimulation parameters. Would survival of BrdU neurons not be compromised by the expression of mutated APP after the end of the stimulation? This needs to be discussed.

Our replies: As suggested by the reviewer, we have now described the TBS protocol in the text of p25-L1-4 as the following; "TBS consisted of 10 trains of stimuli at 10 seconds intervals, with each train containing bursts of 4 spikes at 100 Hz and repeated 10 times at 5 Hz and was applied for 16 consecutive days (16-days TBS) in normal adult mice and for 28 consecutive days (28-days TBS) in AD mice". "A 28-days TBS protocol was used in AD mice because our earlier studies have shown that it effectively intervenes the decays of synaptic transmission and improves spatial memory in AD mice^{18,19}."

The underlying mechanisms of the long-lasting effects for DBS are recently reviewed (Penney and Tsai, The road to restoring neural circuits for the treatment of Alzheimer's disease. Nature 539, 187-512, 2016) and have been now discussed in the text of p19-L14-24 of this revision as the following; "A recent work reveals that stimulation of cortical cells with 40 Hz for only once a day is able to generate a long-lasting inhibition of the A β peptide accumulation in AD mice⁴⁸. Activation of a small group of neural cells in the hippocampus was also found to increase the numbers of dendritic spines and improve the spatial learning and memory in AD mice⁴⁹. Together with our findings^{18,19}, it is plausible that a brief activation of a small population of neurons in the brain can restore cellular health and circuit integrity possibly via the release of transmitters and the growth factors that are required for neuronal cell survival and regeneration. Consistent with this, we have recently found that a brief activation of cortical cells with TBS inactivates cell-death associated protein kinase and hence intercepts the decays of excitatory synaptic transmission in AD mice¹⁹."

21. **Reviewer's comments:** Please check mislabeling for Fig 5g. The references for power analysis are not at all related to the behavior tests. How was then power calculated in these experiments?

Our replies: Sorry for this mislabeling for the original Fig. 5g and this has been now corrected in a new Fig. 5. As suggested by the reviewer, we have now described the power calculation in the text of p30-L20-25 of this revision as the following: "power calculations were performed using G*power software v3.1.9.2 (IDRE Research Technology Group, Los Angeles, USA). Group sizes were estimated based on recent studies^{12, 18,19} and were designed to provide at least 80% power with the following parameters: probability of type I error (α)=0.05, a conservative effect size of 0.25, and 3-8 treatment groups with multiple measurements obtained per replicate".

22. **Reviewer's comments:** The concept that acetylcholine, via M1 muscarinic receptors regulates neurogenesis, is not new (see Neuropharmacology. 2010 May;58(6):921-9. doi: 10.1016/j.neuropharm.2009.12.005. Epub 2009 Dec 22. Agonist-induced restoration of hippocampal neurogenesis and cognitive improvement in a model of cholinergic

denervation. Van Kampen JM1, Eckman CB. this is just one example, there are several other publications). Please make sure to reference these initial works adequately.

Our replies: As suggested by the reviewer, we have now cited the articles that are recommended by the reviewer and another related article in this revision (references 41,42).

23. **Reviewer's comments:** Discussion: last paragraph seems a bit pretentious and doesn't really explain how this strategy could be implemented in AD patients. It mix actual therapeutically approaches with techniques used in basic research. With all the information on this manuscript, I think the discussion could be richer. Please avoid odd sentences.

Our replies: As suggested by the reviewer, we have now deleted this sentence and the similar descriptions throughout the text in this revision.

REVIEWERS' COMMENTS:

Reviewer #1 (Remarks to the Author):

The authors have addressed my comments with appropriate changes to the manuscript. In particular, they have tempered the therapeutic relevance of their findings so that the text is now more cautious and in line with the evidence presented. Overall, this is a very interesting and novel report that is likely to stimulate a lot of additional research relevant to understanding and treating Alzheimer's disease.

Reviewer #2 (Remarks to the Author):

The authors improved the manuscript according to the concerns from reviewers. However, there are still many concerns. I can't recommend the publication of current version. The manuscript is still poorly written and looks immature although the results are excellent.

1) I still found much redundancy and overstatements. Although the authors stated that the manuscript was corrected by 3 English editors, I still recommend that the authors should ask "good" English editor to correct English of some parts in this manuscript.

2) Statistical analyses

(1) I suggested that "the authors should show all of the statistical results". However, almost statistical results are missing. More importantly, the authors did not describe results of statistical analyses. Therefore, I could not clarify whether the conclusion of each figure is correct.

(2) I recommended to perform post hoc Bonferroni's comparisons following two way ANOVA. However, I could not find such description in the methods and figure legends.

(3) For example, in the Result session, the authors stated that "The mRNA and protein expression of the M1 receptor "significantly" decreased in the lysates" (Figure 3g). Did they perform normalization of mRNA and protein expressions and present graphs? Additionally, I found that information of antibody and primers are missing.

(4) I could not see the supplemental figures well. The authors should show them using pdfs, but not ppt.

Reviewer #3 (Remarks to the Author):

The manuscript has been reviewed by the authors and most of the requested modifications seem to have been implemented. The work described here presents an important advancement to the field. It shows that cholinergic neurons innervate newly born neurons and that activation of cholinergic signaling facilitates the survival of new neurons. Pattern separation, which has previously shown to be modulated by the incorporation of new neurons in hippocampal circuits seems to be particularly affected.

The authors show also that activation of cholinergic neurons can rescue the deficit in neurogenesis in AD mouse models as well as corresponding behavioral deficits.

Despite the incorporation of revised figures and new data, the text still has some odd expressions. This is also true for the figure legends.

Our responses to the reviewer's comments

To the reviewer 2:

- 1. The reviewer's comments:** I still found much redundancy and overstatements. Although the authors stated that the manuscript was corrected by 3 English editors, I still recommend that the authors should ask "good" English editor to correct English of some parts in this manuscript.

Our replies: As suggested by the reviewer, we have now invited my colleagues including Dr Micheal Salter at University of Toronto and Dr Mingjie Zhang at University of Sciences and Technology HK to edit the manuscript and correct the redundancy overstatements throughout the text.

- 2. The reviewer's comments:** 2) Statistical analyses: (1) I suggested that "the authors should show all of the statistical results". However, almost statistical results are missing. More importantly, the authors did not describe results of statistical analyses. Therefore, I could not clarify whether the conclusion of each figure is correct. (2) I recommended to perform post hoc Bonferroni's comparisons following two way ANOVA. However, I could not find such description in the methods and figure legends. (3) For example, in the Result session, the authors stated that "The mRNA and protein expression of the M1 receptor "significantly" decreased in the lysates" (Figure 3g). Did they perform normalization of mRNA and protein expressions and present graphs? Additionally, I found that information of antibody and primers are missing.

Our replies: We have originally performed two way ANOVA and Student *t*-tests and represented the statistical results in the Figure legends. As suggested by the reviewer, we have now performed post hoc Bonferroni's comparisons following two-way ANOVA. For the clarity, we have now represented all statistical results in new Supplementary Table 2.

As suggested by the reviewer, we have now represented the averaged data in bar graphs of new Supplementary Fig. 7, showing that the expression of mRNA and protein of M1 receptor in the dNGIs^{M1siRNA+} mice were significantly decreased compared with the dNGIs^{MsiRNA+} mice.

We have listed the antibodies and primers used in this study in the section of Methods (p29-L12-18, p30-L4-8).

- 3. The reviewer's comments:** I could not see the supplemental figures well. The authors should show them using pdfs, but not ppt.

Our replies: As suggested by the reviewer, we have now edited legends of all Supplementary Figures and formatted all supplementary Figures as a PDF file.

To the reviewer 3:

The reviewer's comments: Despite the incorporation of revised figures and new data, the text still has some odd expressions. This is also true for the figure legends.

Our replies: As suggested by the reviewer, the manuscript including all figure legends have been now edited based on the editorial requires and the reviewer's comments by Dr Micheal Salter at University of Toronto, and Dr Mingjie Zhang at University of Sciences and Technology HK.